# AERA-MIP: Emission pathways, remaining budgets and carbon cycle dynamics compatible with 1.5 ºC and 2 ºC global warming stabilization

Yona Silvy[1,2], Thomas L. Frölicher[1,2], Jens Terhaar[1,2,3], Fortunat Joos[1,2], Friedrich A. Burger[1,2], Fabrice Lacroix[1,2], Myles Allen[4,5], Raffaele Bernardello[6], Laurent Bopp[7], Victor Brovkin[8], Jonathan R. Buzan[1,2], Patricia Cadule[7], Martin Dix[9], John Dunne[10], Pierre Friedlingstein[7,11], Goran Georgievski[8], Tomohiro Hajima[12], Stuart Jenkins[4,5], Michio Kawamiya[12], Nancy Y. Kiang[13], Vladimir Lapin[6], Donghyun Lee[5], Paul Lerner[13,14], Nadine Mengis[15], Estela A. Monteiro[15], David Paynter[10], Glen P. Peters[16], Anastasia Romanou[13,14], Jörg Schwinger[17], Sarah Sparrow[18], Eric Stofferahn[10], Jerry Tjiputra[17], Etienne Tourigny[6], and Tilo Ziehn[9]

[1]Climate and Environmental Physics, Physics Institute, University of Bern, Switzerland
[2]Oeschger Centre for Climate Change Research, University of Bern, Switzerland
[3]Woods Hole Oceanographic Institution, Woods Hole, MA, USA
[4]Atmospheric, Oceanic and Planetary Physics, Department of Physics, University of Oxford, Oxford, UK
[5]Environmental Change Institute, School of Geography and the Environment, University of Oxford, Oxford, UK
[6]Barcelona Supercomputing Center, Barcelona, Spain
[7]LMD-IPSL, CNRS, Ecole normale supérieure / PSL, Sorbonne Université, Ecole Polytechnique, Paris, France
[8]Max Planck Institute for Meteorology, Hamburg, Germany
[9]CSIRO Environment, Aspendale, Australia
[10]NOAA/OAR Geophysical Fluid Dynamics Laboratory, Princeton, NJ, USA
[11]Faculty of Environment, Science and Economy, University of Exeter, Exeter, UK
[12]Japan Agency for Marine-Earth Science and Technology, Yokohama, Japan
[13]NASA Goddard Institute for Space Studies, New York, NY, USA
[14]Applied Physics and Applied Mathematics, Columbia University, New York, NY, USA
[15]GEOMAR Helmholtz Centre for Ocean Research Kiel, Kiel, Germany
[16]CICERO Center for International Climate Research, Oslo, Norway
[17]NORCE Norwegian Research Centre and Bjerknes Centre for Climate Research, Bergen, Norway
[18]Oxford e-Research Centre, Department of Engineering Science, University of Oxford, Oxford, UK

**Correspondence:** Yona Silvy (yona.silvy@unibe.ch)

**Abstract.** While international climate policies now focus on limiting global warming to well below 2 °C, or pursuing 1.5 °C level of global warming, the climate modeling community has not provided an experimental design in which all Earth System Models (ESMs) converge and stabilize at the same prescribed global warming levels. This gap hampers accurate estimations based on comprehensive ESMs of the carbon emission pathways and budgets needed to meet such agreed warming levels, and of the associated climate impacts under temperature stabilization. Here, we apply the Adaptive Emission Reduction Approach (AERA) with ESMs to provide such simulations in which all models converge at 1.5 °C and 2.0 °C warming levels by adjusting their emissions over time. These emission-driven simulations provide a wide range of emission pathways and resulting atmospheric $CO_2$ projections for a given warming level, uncovering uncertainty ranges that were previously missing

in the traditional CMIP scenarios with prescribed greenhouse gas concentration pathways. Meeting the 1.5 °C warming level requires a 40 % (full model range: 7 to 76 %) reduction in multi-model mean $CO_2$-forcing equivalent ($CO_2$-fe) emissions from 2025 to 2030, a 98 % (57 to 127 %) reduction from 2025 to 2050, and a stabilization at 1.0 (-1.7 to 2.9) PgC yr$^{-1}$ from 2100 onward after the 1.5 °C global warming level is reached. Meeting the 2.0 °C warming level requires a 47 % (8 to 92 %) reduction in multi-model mean $CO_2$-fe emissions until 2050 and a stabilization at 1.7 (-1.5 to 2.7) PgC yr$^{-1}$ from 2100 onward. The on-average positive emissions under stabilized global temperatures are the result of a decreasing transient climate response to cumulative $CO_2$-fe emissions over time under stabilized global warming. This evolution is consistent with a slightly negative zero emissions commitment - initially assumed zero - and leads to an increase in the post-2025 $CO_2$-fe emission budget by a factor 2.2 (-0.8 to 6.9) by 2150 for the 1.5 °C warming level and a factor 1.4 (0.9 to 2.4) for the 2.0 °C warming level compared to its first estimate in 2025. The median $CO_2$-only carbon budget by 2150, relative to 2020, is 800 $GtCO_2$ for the 1.5 °C warming level and 2250 $GtCO_2$ for the 2.0 °C warming level. These median values exceed the median IPCC AR6 estimates by 60% for the 1.5 °C warming level and 67% for 2.0 °C, respectively. Some of the differences may be explained by the choice of the mitigation scenario for non-$CO_2$ radiative agents. Our simulations highlight shifts in carbon uptake dynamics under stabilized temperature, such as a cessation of the carbon sinks in the North Atlantic and in tropical forests. On the other hand, the Southern Ocean remains a carbon sink over centuries after temperatures stabilize. Overall, this new type of warming level-based emission-driven simulations offers a more coherent assessment across climate models and opens up a wide range of possibilities for studying both the carbon cycle and climate impacts, such as extreme events, under climate stabilization.

# 1 Introduction

Climate goals outlined in international policies, such as the 2015 Paris Agreement (UNFCCC, 2015), primarily focus on global warming levels. The Paris Agreement in particular aims to hold "the increase in the global average temperature to well below 2°C above pre-industrial levels" and to pursue efforts "to limit the temperature increase to 1.5°C above pre-industrial levels". Global warming levels are chosen in international policies as they are often directly correlated to global and regional impacts of climate change (IPCC, 2018; Seneviratne et al., 2016). Hence, each fraction of avoided warming reduces risks for humans and ecosystems (IPCC, 2022).

The Coupled Model Intercomparison Project (CMIP) provides climate projections of Earth System Models (ESM) for the 21$^{th}$ century and beyond. These projections, however, follow an approach that poses challenges in estimating carbon emission pathways and budgets that are consistent with the goals of the Paris Agreement. In CMIP projections, ESMs have traditionally been driven by prescribed pathways in the concentrations of $CO_2$ and other radiative agents (O'Neill et al., 2016; Meinshausen et al., 2020), although there is now a push towards more emission-driven scenario designs (Sanderson et al., 2023). For a given greenhouse gas emissions or concentrations trajectory, each ESM simulates different global warming trajectories (e.g., Tebaldi et al., 2021, and see schematic in Fig.1), primarily due to the wide range in climate sensitivity and in the transient climate response (e.g. Zelinka et al., 2020; Meehl et al., 2020; Arora et al., 2020). The varying responses of ESMs have lead to varying estimates for the cumulative $CO_2$ emissions until a given global warming level is reached (Rogelj et al., 2016; Tokarska

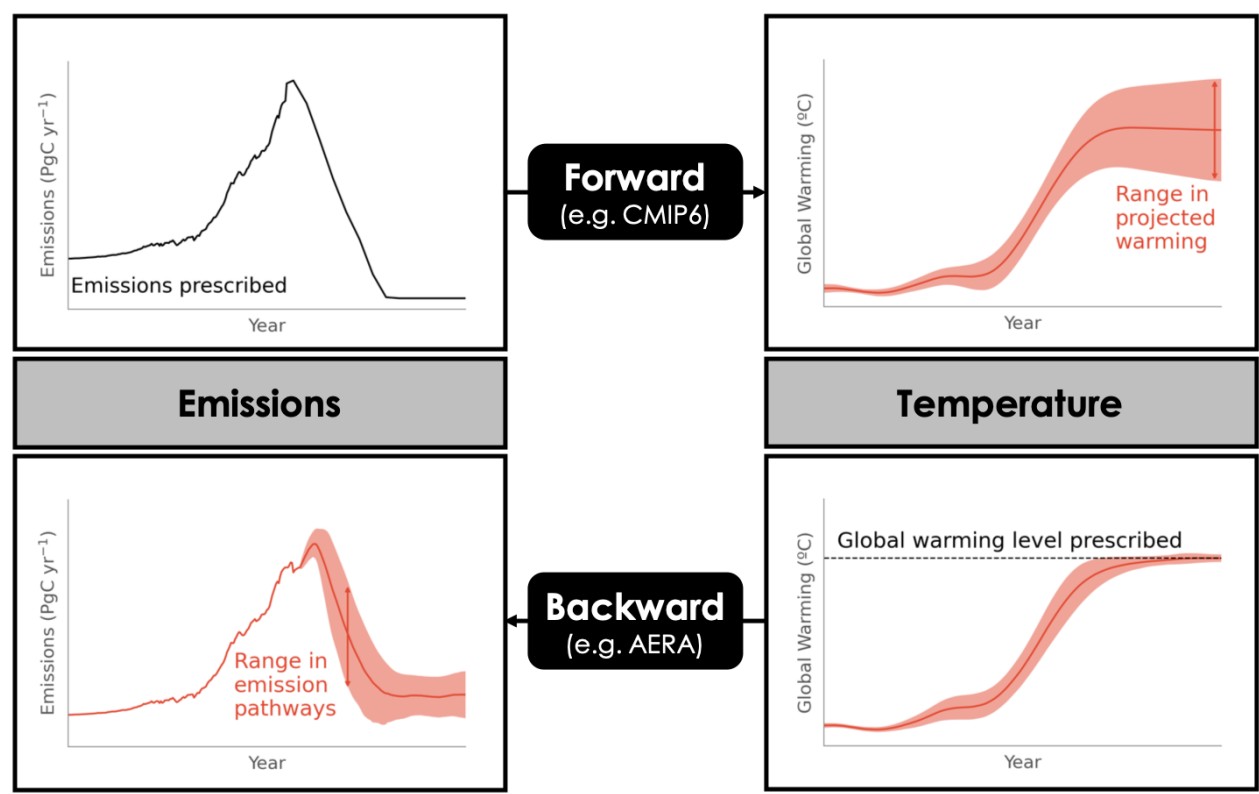

**Figure 1.** Schematic of the forward vs. backward modelling approach. Black lines indicate the same prescribed variable (emissions for CMIP, global warming level for AERA) in all models. Red lines indicate the resulting range across models in simulated temperature, and the resolved range in emissions in AERA.

et al., 2018). Consequently, it is difficult to estimate the emission pathways that align with stabilizing the climate at specific global warming levels. Instead, the emission pathways and budgets for temperature stabilization were estimated with reduced-form models (Millar et al., 2017), Earth System Models of Intermediate Complexity (Steinacher et al., 2013; Matthews et al., 2017; Goodwin et al., 2018b; Mengis and Matthews, 2020; Matthews et al., 2021), or by the near-linear relationship between anthropogenic global surface warming and cumulative emissions suggested by ESMs (Allen et al., 2009; Matthews et al., 2009; Zickfeld et al., 2009).

Numerous studies have confirmed the near-linear response between global temperature increase and cumulative $CO_2$ emissions, with the slope of this relationship called the Transient Climate Response to Cumulative $CO_2$ emissions (TCRE). The TCRE has been estimated from observational data (e.g. Millar and Friedlingstein, 2018) and model experiments, and has been used to estimate the amount of cumulative emissions that could still be emitted before reaching a given global warming level (Meinshausen et al., 2009; Rogelj et al., 2016, 2019). The amount of cumulative $CO_2$ emissions up to the point of net zero $CO_2$

emissions is known as the remaining carbon budget (RCB) for a fixed warming level. The RCB from the beginning of year 2020, for example, has been estimated to be 400 GtCO$_2$ (109 PgC) for a 50 % chance of maintaining the global temperature below 1.5 ℃ of warming, with its uncertainty range being 54 to 204 PgC, for an 83 %-17 % chance (Forster et al., 2023). For the 2.0 ℃ level, the RCB ranges from 245 to 600 PgC. The uncertainty of the RCB is large due to uncertainties in the TCRE, climate response to zero emissions, unrepresented feedbacks, future warming from non-CO$_2$ agents (Rogelj et al., 2019), and pathway dependencies (Millar et al., 2017).

As the uncertainties in the TCRE, feedbacks, and future warming from non-CO$_2$ agents will unlikely be significantly reduced in the near future, an iterative and regular update of the RCB is necessary to ensure that the latest scientific understanding is included and to ensure policies can be judiciously implemented to avoid exceeding the given global warming levels. Through an adaptive process aligned with the "pledge and review" mechanism of the Paris Agreement, the RCB and emission pathways might be regularly updated at each stocktake period based on the best available science (Otto et al., 2015). An iterative approach ensures that the emission pathway remains in line with the prescribed warming level and adaptively adjusts to uncertainties in the evolution of the climate response, such as the potential warming or cooling at near-zero CO$_2$ emissions (Zero Emissions Commitment - ZEC - MacDougall et al., 2020), and the response to mitigation rates in non-CO$_2$ radiative agents. Such adaptive approaches have been proposed and tested with reduced complexity models running forward from present day, and offer promising potential. Goodwin et al. (2018a) introduced the "Adjusting Mitigation Pathways" method using a climate box model. In this approach, the remaining carbon budget to a predefined warming level is estimated by using the near-linear relationship between warming and cumulative carbon emission from a historically forced simulation. Then the remaining budget is distributed in the future and reassessed every ten years. However, limitations, including idealized emission trajectories, disregarding non-CO$_2$ agents in TCRE and RCB calculations, and challenges in reaching the warming level within the uncertainty range of $\pm$ 0.25 ℃ by 2300, hinder its real-world applicability. Further approaches which do not make use of an adaptive RCB include backwards modelling approaches, like temperature tracking using models of intermediate complexity (EMICs; Matthews and Caldeira (2008); Zickfeld et al. (2009, 2013); Mengis et al. (2018)) or employing impulse-response functions with a simple climate model (Millar et al., 2017) to convert a smooth temperature trajectory into an emission pathway. In contrast, the recently developed Adaptive Emission Reduction Approach (AERA), validated with the Bern3D-LPX Earth System Model of Intermediate Complexity (Terhaar et al., 2022a, 2023), offers a less idealized approach and addresses several limitations inherent in previous approaches. Even though the underlying principles of AERA are similar to the Adjusting Mitigation pathways approach (Goodwin et al., 2018a), AERA provides smoother emission pathways, incorporates non-CO$_2$ agents, always stabilizes at the warming levels within $\pm$0.2 ℃, and can also be applied to run simulations that temporarily overshoot the warming level. However, as of now, none of these approaches have been implemented in comprehensive fully coupled ESMs.

Here, we implement the AERA introduced by Terhaar et al. (2022a) across a range of fully-coupled CO$_2$ emission-driven Earth System Models. This new Adaptive Emission Reduction Approach Model Intercomparison Project (AERA-MIP) provides projections that stabilize surface temperature to the same warming level. The AERA-MIP framework enables estimations of the remaining carbon budget, compatible emission pathways, and the ocean and land carbon cycle response to those path-

ways within scenarios of stabilized warming levels at 1.5 ℃ or 2 ℃, which we present in this study. Other potential applications of AERA-MIP will be addressed in the Conclusions.

## 2   Methods

The detailed protocol for the simulations of the AERA-MIP is provided in Frölicher et al. (2022). The protocol is a slightly modified version of the originally proposed method by Terhaar et al. (2022a). For details, see steps 1 and 3 in section 2.1. The AERA code is distributed as a python module openly available under https://github.com/Jete90/AERA, with a guided documentation and examples. Below, the protocol is summarized before introducing the participating models, simulations, and carbon stock analysis.

### 2.1   Adaptive emission reduction approach - AERA

The goal of the AERA is to quantify successive future trajectories of $CO_2$-forcing equivalent ($CO_2$-fe) emissions that stabilize the global surface air temperature (GSAT) at a predetermined level (Terhaar et al., 2022a). The concept of $CO_2$-fe emissions is used to unify the emissions of various radiative forcing agents and precursors into a single metric (Jenkins et al., 2018; Allen et al., 2018; Smith et al., 2021). $CO_2$-fe emissions for all non-$CO_2$ agents represent the $CO_2$ emissions that would produce the same radiative forcing trajectory as these non-$CO_2$ emissions. While cumulative $CO_2$ emissions largely determine anthropogenic warming, non-$CO_2$ radiative forcing agents such as methane, nitrous oxide and aerosols also play an important role. Quantifying the impact of these non-$CO_2$ agents on global temperatures is complicated by existing methodologies, which often use conventional Global Warming Potentials or other metrics to convert the non-$CO_2$ radiative forcing agents into '$CO_2$-equivalent' emissions. The $CO_2$-fe emissions framework offers an alternative, well-suited for comparing emissions from different agents in the context of temperature stabilization pathways (Terhaar et al., 2022a). It also offers an opportunity to compare emission reductions of different radiative forcing agents for ecosystem impacts (Terhaar et al., 2023). The $CO_2$-fe emissions from non-$CO_2$ agents, $E_{non\text{-}CO_2\text{-}fe}$, are estimated based on the radiative forcing time series of non-$CO_2$ agents (Smith et al., 2021):

$$E_{non\text{-}CO_2\text{-}fe}(t) = \frac{1}{\alpha}\left(\frac{\mathrm{d}F_{non\text{-}CO_2\text{-}fe}(t)}{\mathrm{d}t} + \rho F_{non\text{-}CO_2\text{-}fe}(t)\right) \tag{1}$$

where $F_{non-CO_2}$ is the radiative forcing of non-$CO_2$ agents, $\rho$ is the rate of decline in radiative forcing under zero emission over decadal to centennial timescales (0.33%), and $\alpha$ is a constant representing the forcing impact of ongoing $CO_2$ emissions (1.08 Wm$^{-2}$ per 1000 GtC). By varying the relative shares of $CH_4$ and $N_2O$ emissions and radiative forcing from aerosols in the total $CO_2$-fe emissions, Terhaar et al. (2022a) demonstrated the robustness of the $CO_2$-fe approach in translating contributions from different radiative forcing agents into $CO_2$-fe emissions.

AERA achieves the temperature stabilization at a predetermined warming level by estimating future $CO_2$-fe emissions based on the estimated TCRE-fe (the transient climate response to cumulative $CO_2$-fe emissions). The TCRE-fe in turn is derived

from simulations of past annual GSAT, past fossil fuel and land-use change $CO_2$ emissions, and $CO_2$-fe emissions from non-$CO_2$ forcing agents calculated from their radiative forcing estimates (Jenkins et al., 2021).

The AERA consists of three steps, which are repeated every five years, mirroring the stocktaking mechanism implemented in the Paris Agreement (Terhaar et al., 2022a).

1. AERA estimates the anthropogenic warming $\Delta T$ from the simulated GSAT time series (relative to 1850-1900). The past anthropogenic warming is here estimated by applying a 31-year running mean with a linear extrapolation for the last 15 years, assuming constant warming rate based on the last 31 years. This is in contrast to the original AERA method, which employs an impulse response function on radiative forcing and temperature estimates to determine anthropogenic warming (Otto et al., 2015). The simple running mean method was applied here as it yields more robust results in cases where the model's radiative forcing is unknown (as is the case for most ESMs), and after reaching the temperature level.

2. The anthropogenic warming calculated in step 1 is then divided by the cumulative $CO_2$-fe emissions since 1850 to determine the TCRE-fe. Using this TCRE-fe metric, we compute the amount of $CO_2$-fe emissions that can still be emitted before reaching the temperature level. This remaining $CO_2$-fe emission budget, referred to as REB, is derived by dividing the remaining warming until the temperature level is reached ($\Delta T_{remaining}$) by the TCRE-fe value:

$$REB = \frac{\Delta T_{remaining}}{TCRE\text{-}fe} \tag{2}$$

3. The REB from step 2 is then distributed in the future using a cubic polynomial function. The parameters of the function are chosen to limit an overshoot in temperature and maintain minimal year-to-year changes in $CO_2$-fe emissions (see Terhaar et al. (2022a) for details). In contrast to Terhaar et al. (2022a) and to prevent large oscillations in the emissions in the ESMs when the warming levels are reached, we modified the minimum and maximum length (both now variable and depending on the REB and the annual emissions) of the cubic polynomial that distributes the $CO_2$-fe emissions over the future years.

## 2.2 Earth System Models

Thirteen models have participated in the AERA-MIP: ten fully coupled Earth System Models (ACCESS-ESM1-5, CESM2, EC-Earth3-CC, NASA-GISS-E2-1-G-CC, GFDL-ESM2M, GFDL-ESM4, IPSL-CM6-LR-ESMCO2, MIROC-ES2L, MPI-ESM1-2-LR and NorESM2-LM), two models of intermediate complexity (Bern3D-LPX and UVic-ESCM-2.10), and one atmosphere-ocean general circulation model coupled to a carbon cycle emulator (HadCM3-FaIR2, see Appendix A for a description of the configuration). Nine of the fully-coupled ESMs have participated in the sixth phase of the coupled model intercomparison project (CMIP6), and GFDL-ESM2M has participated in CMIP5. Table A1 lists the models, their abbreviation used in this paper, as well as the corresponding references, the simulated time period and the number of ensemble members. A summary of their components has been already provided in several multi-ESM studies (e.g., Séférian et al., 2020; MacDougall et al., 2020; Arora et al., 2020; Canadell et al., 2021).

Small initial condition ensembles were provided by two participating ESMs to estimate the uncertainty associated with internal variability (Table A1). EC-Earth provided a three-member and GFDL-ESM2M a five-member ensemble. Ideally, more ensemble members are necessary to properly quantify the internal variability (Lehner et al., 2020). However, even the small number of ensemble members available here provide a first-order estimate of internal variability.

The ACCESS-ESM1-5 model is somewhat an outlier. It also converges to both prescribed temperature levels, but substantially later than all other models. The model most likely converges later due to a strong mismatch between the estimate of non-$CO_2$ radiative forcing used in the AERA to estimate $CO_2$-fe emissions (from the RCP/SSP database, see below) and the non-$CO_2$ radiative forcing simulated in the model based on the prescribed atmospheric non-$CO_2$ radiative agents. Hence, results from ACCESS were excluded from subsequent multi-model statistics but are still shown for transparency in Fig. 2. However, the inclusion of the ACCESS model in the ensemble modifies our results only slightly (Fig. B1).

## 2.3 Simulations

The simulations of the AERA-MIP are performed until at least 2150 (except for HadCM3-FaIR2 until 2100) and up to 2300 to allow for enough time to reach the temperature level and stabilize global surface warming (Table A1). Simulations have been conducted for both the 1.5 °C and 2.0 °C temperature levels. To remove biases in simulated warming over the historical period relative to observations, we use the concept of a relative temperature level (Millar et al., 2017; Goodwin et al., 2018a; Terhaar et al., 2022a). Under this concept, the remaining allowable warming in 2020 is first estimated from observations. In a second step, the AERA adds this observation-based remaining allowable warming to the models' anthropogenic warming in 2020 to calculate the absolute temperature level in each model (see Frölicher et al., 2022; Terhaar et al., 2022a). Thus, each model estimates the emission trajectory for the same remaining allowable warming in 2020, which is here estimated to be 0.28 ℃ for the 1.5 ℃ warming level and 0.78 °C for the 2.0 ℃ level. These values were derived from the observation-based estimated warming of 1.22 °C in 2020 (Terhaar et al., 2022a). Each initial condition ensemble member has its own anthropogenic temperature in year 2020, resulting in very small differences in absolute temperatures for that year (maximum differences across GFDL ESM2M ensemble members of 0.067°C).

Following a similar simulation strategy as in Terhaar et al. (2022a), all simulations branch off an emission-driven simulation (*esm-hist*) over the historical period following the CMIP6 protocol. After the end of the CMIP6 historical period in 2014, fossil fuel $CO_2$ emissions follow observed emissions until 2020 (Friedlingstein et al., 2020; Le Quéré et al., 2021) and projected emissions from the Nationally Determined Contributions (NDCs; Climate Action Tracker: https://climateactiontracker.org/global/temperatures/, last accessed December 2021) from 2021 to 2025. Starting at the end of year 2025, fossil-fuel $CO_2$ emissions prescribed to the model are obtained every five years from AERA (see below and schematic in Fig. B2) by subtracting $CO_2$-fe emissions from prescribed non-$CO_2$ radiative agents and land use land cover change from the AERA-derived total $CO_2$-fe emission curve. Non-$CO_2$ agents as well as land use and land cover change are prescribed as in the SSP1-2.6 scenario after 2015. This low emission high mitigation scenario is often used in related climate stabilization approaches (e.g. Millar et al. (2017); Mengis et al. (2018)). An exception is the CMIP5-type GFDL-ESM2M which follows the CMIP5 protocol for the historical period as well as the RCP2.6 scenario instead of the SSP1-2.6 scenario for the non-$CO_2$ forcing agents and land use

and land cover change. After 2100, no further changes occur in non-$CO_2$ forcing agents, land use and land cover. A schematic is provided in Fig. B2, summarising the feedback loop between the AERA module and the ESM during the post-2025 period of the simulations.

As the AERA calculates the total $CO_2$-fe emissions, it requires information about emissions from non-$CO_2$ forcing agents ($E_{non\text{-}CO_2\text{-}fe}$) and $CO_2$ emissions from land use and land cover change ($E_{LUC}$; referred to as land use change emissions for simplicity from now on) to estimate the past TCRE-fe, on top of the past fossil fuel $CO_2$ emissions ($E_{FOS}$). Moreover, the future $E_{FOS}$ prescribed to the model after each stocktake are calculated as the difference between the $CO_2$-fe emission curve ($E_{total\text{-}fe}$) from Step 3 of AERA, and the estimated future $CO_2$-fe emissions from land use change and non-$CO_2$ forcing agents:

$$E_{FOS} = E_{total\text{-}fe} - E_{LUC} - E_{non\text{-}CO_2\text{-}fe} \tag{3}$$

As most models do not directly output the radiative forcing of non-$CO_2$ agents (required to derive $E_{non\text{-}CO_2\text{-}fe}$), we estimate this time series from the radiative forcing given by the RCP/SSP database for both the historical period and SSP1-2.6 scenario. Some models, however, have provided an estimate of the simulated effective radiative forcing for all non-$CO_2$ radiative agents (see Table A2, and Fig. 4d). For these models, the internally calculated effective radiative forcing estimates were used to derive the $CO_2$-fe emissions from the non-$CO_2$ forcing agents (Smith et al., 2021). Similarly, some models (see Table A2 and Fig. 4c) have conducted additional simulations to estimate $E_{LUC}$ (Lawrence et al., 2016; Liddicoat et al., 2021). In that case, the model-specific $E_{LUC}$ is prescribed to AERA instead of the default $E_{LUC}$ estimate stemming from the Bern3D-LPX model that was scaled to align with the Global Carbon Budget estimates from 1850 to 2020. The default $E_{non\text{-}CO_2\text{-}fe}$ and $E_{LUC}$ time series as well as the model-specific emissions for those who were able to estimate them are shown in Figs. 4 and A1.

Although the future $E_{LUC}$ and $E_{non\text{-}CO_2\text{-}fe}$ are prescribed in AERA to enable the extraction of future $E_{FOS}$ as input to the models (Fig. B2), the future $CO_2$-fe emission curve $E_{total\text{-}fe}$ is largely insensitive to the chosen land use and non-$CO_2$ forcings (see tests in Terhaar et al. (2022a)).

Further details on the configuration of the AERA simulations are provided in Appendix A.

## 2.4 Remaining emission budget over time

While the TCRE is commonly considered approximately constant, recent studies suggest a potential variability in TCRE when $CO_2$ emissions are reduced, owing partly to a non-zero ZEC (Frölicher and Paynter, 2015; Steinacher and Joos, 2016; Nicholls et al., 2020; MacDougall et al., 2020). This indicates that the REB originally estimated at the end of 2025 may evolve over time, with the additional effect of potential non-linearities in the response of non-$CO_2$ forcing agents. To illustrate the temporal evolution of the REB from a fixed starting point, we reconstruct the $CO_2$-fe emission budget from beginning of year 2026 ($EB_{2026}$) at each stocktake by summing the REB (i.e. all future emissions) calculated at the end of the stocktake year ($t_{st}$) and the already emitted $CO_2$-fe emissions between 2026 and that year:

$$EB_{2026}(t_{st}) = REB(t_{st}) + \int_{2026}^{t_{st}} E_{total\text{-}fe}(t)dt \tag{4}$$

$EB_{2026}(2025)$ is by definition $REB(2025)$.

## 2.5 Distribution of carbon in the Earth System

Within individual ESMs, all sources and sinks of the carbon mass balance are known and we can write:

$$E_{FOS} = G_{ATM} + S_{OCEAN} + (S_{LAND} - E_{LUC}) \tag{5}$$

$E_{FOS}$ are the prescribed global fossil fuel $CO_2$ emissions during the historical period and calculated by AERA post 2025. $G_{ATM}$ indicates the simulated atmospheric $CO_2$ growth rate in PgC yr$^{-1}$ by using the conversion factor of 2.123 PgC ppm$^{-1}$ (Enting et al., 1994). $S_{OCEAN}$ is the net ocean carbon sink derived from the $CO_2$ flux into the ocean (CMIP6 variable *fgco2*). $S_{LAND}$-$E_{LUC}$ denotes the net $CO_2$ flux into land and is derived from the net biosphere production (CMIP6 variable *nbp*). The net biosphere production includes the gross land carbon sink $S_{LAND}$ minus emissions from land use and land cover change, $E_{LUC}$.

For the eight models that estimate their $E_{LUC}$ term, we can quantify the gross land carbon sink $S_{LAND}$ (Figure A1 and Appendix A), by adding the two diagnostics *nbp* + $E_{LUC}$. Thus, by rearranging Equation 5, we can separate net sources and sinks of $CO_2$:

$$E_{FOS} + E_{LUC} = G_{ATM} + S_{OCEAN} + S_{LAND} \tag{6}$$

As opposed to earlier studies (Liddicoat et al., 2021; Koven et al., 2022), atmospheric $CO_2$ is freely evolving and not prescribed, whereas $E_{FOS}$ is prescribed and does not need to be diagnosed.

From Equation 6, we derive the cumulative airborne ($C_{AF}$), ocean-borne fraction ($C_{OF}$) and land-borne fraction ($C_{LF}$):

$$C_{AF}(t) = \frac{\int G_{ATM}(t)dt}{\int (G_{ATM}(t) + S_{OCEAN}(t) + S_{LAND}(t))dt} \tag{7}$$

$$C_{OF}(t) = \frac{\int S_{OCEAN}(t)dt}{\int (G_{ATM}(t) + S_{OCEAN}(t) + S_{LAND})(t))dt} \tag{8}$$

$$C_{LF}(t) = \frac{\int S_{LAND}(t)dt}{\int (G_{ATM}(t) + S_{OCEAN}(t) + S_{LAND})(t))dt} \tag{9}$$

with the $\int$ sign representing the time integral from 1850. Here we use $G_{ATM} + S_{OCEAN} + S_{LAND}$ instead of $E_{FOS} + E_{LUC}$ as the denominator of the cumulative fractions to avoid minor budget closure errors. These errors are negligible in most models, represent maximum a few percents (not shown), in which case they are very likely due to missing diagnostics.

For some models, the $E_{LUC}$ term prescribed to AERA was either the default estimate from the adjusted Bern3D-LPX time series, or presented some errors. In the carbon stock analysis, we corrected these terms (see Appendix A).

## 3 Temperature, $CO_2$-fe emission and atmospheric $CO_2$ pathways

### 3.1 Convergence towards the prescribed temperature level

The AERA effectively stabilizes the simulated GSAT in ESMs around the prescribed warming level within an uncertainty of ±0.2 ℃ (Fig. 2a,b). The uncertainty of ±0.2 ℃ corresponds to the uncertainty with which anthropogenic warming can be
determined from observations (Haustein et al., 2017; Jenkins et al., 2022a). The IPSL model temporarily leaves the 1.5 °C uncertainty range, and MIROC briefly leaves the 2.0 °C uncertainty range. In the 1.5 °C simulation, the multi-model mean GSAT anomaly enters the warming level uncertainty range in 2026, i.e., the year when the first AERA period begins (black thick line in Fig. 2a). The temperature anomaly first peaks at 1.43 ℃ in 2043 (min-max model range: 1.20 ℃ to 1.57 ℃), temporarily drops to 1.40 ℃ (1.23 ℃ to 1.50 ℃) in 2069, and stabilizes around 1.44 ℃ (1.25 ℃ to 1.64 ℃) between 2100 and
2150. In the 2.0 °C simulation (Fig. 2b), the multi-model mean GSAT anomaly enters the uncertainty envelope in 2061, and stabilizes at around 1.90 ℃ (1.77 ℃ to 2.09 ℃) between 2100 and 2150.

The convergence to the temperature level here shows that the AERA approach works for both intermediate complexity models, as shown previously (Terhaar et al., 2022a), as well as for fully-coupled ESMs. This is the case despite differences in $E_{LUC}$ and $E_{non\text{-}CO_2\text{-}fe}$ prescribed to the AERA framework and in the models themselves. An exception is the ACCESS
model that only converges to the respective warming level by the late 22$^{nd}$ century after an overshoot of 0.3-0.5 °C (larger overshoot for the 1.5 °C warming level; see Methods for more details).

### 3.2 Compatible $CO_2$-fe emission pathways

For the 1.5 ℃ warming level, $CO_2$-fe emissions decrease strongly and immediately after 2025 for all models, albeit with a large inter-model spread (Fig. 2c; Table 1). By 2030, $CO_2$-fe emissions drop to 8.1 (min-max range: 3.1 to 11.9) PgC yr$^{-1}$, a 40 % (7
% to 76 %) decline compared to 2025 levels of 13.7 (11.0 to 18.9) PgC yr$^{-1}$. During this strong decline phase, $CO_2$-fe emissions reach a maximum reduction of -2.0 (-0.4 to -3.6) PgC yr$^{-2}$. Afterwards, the emissions reach a temporary minimum at nearly zero in 2050 (0.2 PgC yr$^{-1}$; range: -3.6 to 5.0 PgC yr$^{-1}$), corresponding to a 98 % (57 % to 127 %) decline from 2025 levels, before peaking at 1.9 (-0.2 to 4.2) PgC yr$^{-1}$ in 2077. This bounce in the $CO_2$-fe emission curve could be explained by the very rapid mitigation of non-$CO_2$ forcing agents in the early decades of the SSP1-2.6 scenario, causing slightly negative forcing-
equivalent emissions approximately from 2030 to 2100 ($E_{non\text{-}CO_2\text{-}fe}$, see Fig. 4d). Because this decline in $E_{non\text{-}CO_2\text{-}fe}$ is not accounted for in the first estimate of the REB at the end of year 2025, the emission budget is re-evaluated and increases in the stocktake years around 2050 relative to its 2025 estimate (Fig. 5a), leading to the on-average increase in emissions between 2050 and 2077. Subsequently, $CO_2$-fe emissions decrease again to 1.0 (-1.7 to 2.9) PgC yr$^{-1}$ between 2100 and 2150 when global mean temperatures have been stabilized. Until the end of the 22$^{nd}$ century, $CO_2$-fe emissions remain slightly positive
on average. The positive emissions and large model spread during the temperature stabilization phase are consistent with the overall negative but highly uncertain multi-decade temperature response after zero $CO_2$ emissions across a range of EMICs and ESMs (MacDougall et al., 2020; Jenkins et al., 2022b). In 4 out of 13 models (GFDL-ESM4, IPSL, MPI, UVic), negative $CO_2$-fe emissions are not necessary to stabilize at the 1.5 °C warming level.

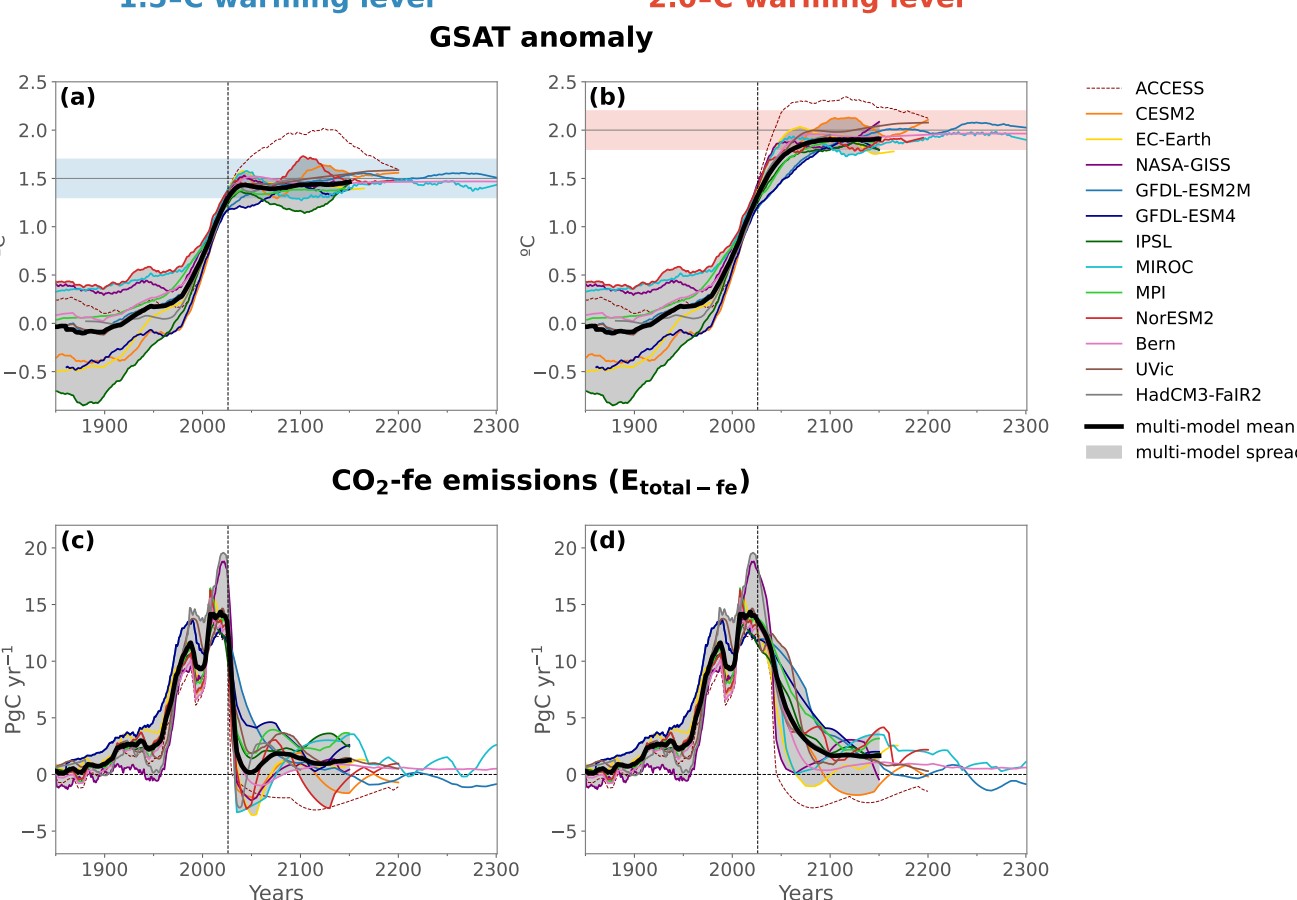

**1.5ºC warming level**      **2.0ºC warming level**

**GSAT anomaly**

**CO$_2$-fe emissions (E$_{total-fe}$)**

**Figure 2.** Simulated temperature anomaly and CO$_2$-fe emissions for the 1.5 °C and 2.0 °C warming levels. Panels a) and b) display the 31-year running mean of the global surface air temperature (GSAT) anomaly, aligned with the observed value in 2020. Panels c) and d) illustrate CO$_2$-fe emissions. The multi-model mean, excluding the ACCESS model, is displayed by the black thick line, with the grey shading indicating the min-max spread. The ensemble mean is shown for models that have several ensemble members. The vertical dotted line at year 2026 marks the beginning of the AERA simulations. The horizontal shading in (a) and (b) indicates the uncertainty with which anthropogenic warming can be determined ($\pm$ 0.2 °C).

Achieving the 2.0 ºC warming level also requires strong CO$_2$-fe emission reductions, albeit less drastic than for the 1.5 ºC level. By 2030, multi-model mean CO$_2$-fe emissions decrease to 13.0 (10.9 to 17.1) PgC yr$^{-1}$, a 5.2 % (-1 % to 16 %) reduction from 2025 levels. By 2050, they further drop to 7.2 (1.3 to 11.4) PgC yr$^{-1}$, a 47 % (8 % to 92 %) decrease compared to 2025, and stabilize at 1.7 (range: -1.5 to 2.7) PgC yr$^{-1}$ between 2100 and 2150. The maximum reduction rate between 2026 and 2100 is -0.7 (range: -2.3 to -0.1) PgC yr$^{-2}$. Only two models, EC-Earth and CESM2, exhibit temporary negative CO$_2$-fe emissions before 2150.

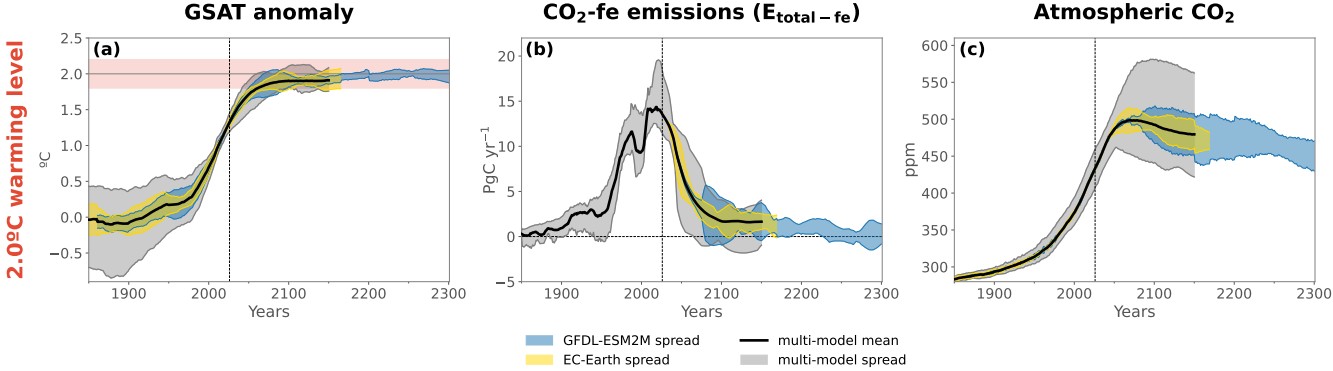

**Figure 3.** Comparison between model uncertainty and a lower bound for internal variability uncertainty in simulated temperature, $CO_2$-fe emissions and atmospheric $CO_2$ pathways for the 2.0 °C warming level. Similar to Fig. 2b,d, but the panels show the multi-model min-max range in grey, and the inter-member min-max range of EC-Earth (3 members) in yellow and GFDL-ESM2M (5 members) in blue. The inter-member ensemble ranges are centered on the multi-model mean. For the 1.5 °C warming level, refer to Fig. B3.

Several models simulate decadal-scale oscillations in emissions trajectories after reaching the prescribed temperature level. These fluctuations partly stem from the challenge faced by AERA's anthropogenic warming estimation, utilizing an extended 31-year running mean, in distinguishing entirely between multi-decadal internal variability and anthropogenic trends in atmospheric temperature. In particular in NorESM2, large multi-decadal temperature variations due to Atlantic Meridional Overturning Circulation (AMOC) decline and subsequent recovery have been shown to occur when emissions are phased
out (Schwinger et al., 2022). However, these oscillations lead to temperature fluctuations that remain within the $\pm\,0.2°C$ range around the temperature level, except for a small undershoot simulated in the IPSL model, which takes a few decades to recover towards the warming level. This slow evolution of GSAT could be partly related to the large low-frequency internal variability exhibited in this model (Bonnet et al., 2021).

     Beyond 2200, a time period for which only three models provide results, $CO_2$-fe emissions necessary for maintaining
temperature stabilization are still projected to evolve slightly. For example, GFDL-ESM2M shows a slower reduction in $CO_2$-fe emissions during the 21st century compared to other models due to its low TCRE and a negative temperature response to zero emissions on decadal timescales. However, by the mid-22nd to 23rd century, $CO_2$-fe emissions in this model become slightly negative and will remain so for several centuries in response to a time-varying ZEC and continued warming in this model on multi-centennial timescales under zero emissions (Frölicher et al., 2014; Frölicher and Paynter, 2015). On the other hand,
the Bern and MIROC models simulate continuous (albeit small) positive emissions on these timescales, reflecting continued cooling in these models under zero $CO_2$ emissions (negative ZEC) on multi-centennial timescales (MacDougall et al., 2020). This time dependency of the ZEC response following a period of $CO_2$ emissions erodes the value of the remaining budget concept as a fixed estimate beyond the first several decades around net zero.

     The simulated uncertainty range in $CO_2$-fe emission pathways across the models is not solely caused by model differences
but also by internal variability (as diagnosed by the range across individual ensemble members of one model; Table 1, Figs.

3b and B3b). The internal variability uncertainty range diagnosed here represents a lower bound of the true internal variability uncertainty from ESMs as only a few ensemble members are available. Across the 5 ensemble members of GFDL-ESM2M, the min-max range in $CO_2$-fe emissions in 2050 is 3.9 PgC yr$^{-1}$ for the 1.5 °C level and 0.7 PgC yr$^{-1}$ for the 2.0 °C level, corresponding to 45 % and 7 % of the multi-model range, respectively. These numbers rise to 66 % and 53 % between 2100 and 2150, respectively. The EC-Earth 3-member ensemble represents 33 % and 53 % of the multi-model range $CO_2$-fe emissions in 2050 for the 1.5 ºC and 2.0 ºC levels, and respectively 34 % and 29% between 2100 and 2150. The inter-member spread decreases during the temperature stabilization period. For example, in the 2.0 ºC warming level ensemble by GFDL-ESM2M, the inter-member range decreases from 2.7 PgC yr$^{-1}$ in 2100-2150 to 1.1 PgC yr$^{-1}$ in 2250-2300 (Fig. 3b). As for the maximum reduction rate in $CO_2$-fe emissions between 2026 and 2100, the GFDL-ESM2M ensemble members range from -1.3 to -0.1 PgC yr$^{-2}$ for the 2.0 °C warming level, corresponding to 56 % of the spread across all models.

### 3.3   Emissions from fossil fuel, non-$CO_2$ agents and land use change

Fossil fuel emissions ($E_{FOS}$, see Equation 3) closely track the evolution of the $CO_2$-fe curve (Fig. 4a,b). In the 1.5 ºC scenario, the multi-model mean emissions remain slightly positive throughout the simulation, reaching a minimum in 2046 of 0.9 (-2.7 to 6.6) PgC yr$^{-1}$, peaking in 2069 at 1.7 (-1.0 to 5.4) PgC yr$^{-1}$, and gradually declining towards zero by the end of the 22$^{nd}$ century. Although the multi-model mean suggests that $E_{FOS}$ can remain positive if emission reductions are fast and strong, negative emissions may still be necessary as five models simulate negative $E_{FOS}$, reaching -4.5 to -1.6 PgC yr$^{-1}$ before 2050.

In the 2.0 ºC simulation, the multi-model mean reaches 8.2 (1.4 to 11.9) PgC yr$^{-1}$ in 2050, stabilizes at 1.3 (-1.7 to 3.2) PgC yr$^{-1}$ between 2100 and 2150, and slightly declines thereafter. In this scenario, only two models (CESM2 and EC-Earth) simulate non-negligible negative fossil fuel emissions before 2150, reaching -1.5 to -2.0 PgC yr$^{-1}$. A temporary increase in $E_{FOS}$ occurs post 2025 in the 2.0 ºC simulation due to rapid reductions in non-$CO_2$ emissions ($E_{non-CO_2-fe}$) as prescribed in the SSP1-2.6 scenario (Fig. 4d).

The prescribed $CO_2$-fe emissions from non-$CO_2$ agents based on SSP1-2.6 (Fig. 4d; colored lines for models that were able to estimate their internal radiative forcing; black line for the others; see Methods) exhibit a rapid decrease after year 2026, reaching maximum negative emissions around year 2054 of -1.4 PgC yr$^{-1}$ and stabilizing at slightly positive levels of 0.2 PgC yr$^{-1}$ after 2100 for the default prescribed emissions (black line). The simulated $CO_2$ emissions based on prescribed land use change from SSP1-2.6 (Fig. 4c) generally remain slightly positive throughout the 21$^{st}$ century, except in the NASA-GISS, EC-Earth and ACCESS models, where they are temporarily negative. Post-2100, the land-use change $CO_2$ emissions prescribed to AERA stabilize around zero, consistent with the constant land use area forcing. For the UVic and MPI simulations, the values prescribed to AERA post-2100 in $E_{LUC}$ were kept positive and constant, although this does not correspond to the land use forcing. These time series were corrected in the remaining emissions budgets reported in Section 4 and in the carbon stock analysis of Section 6 (see Appendix A). Both the non-$CO_2$ and the land-use change $CO_2$-fe emissions, follow identical trajectories for the 1.5 °C and 2.0 °C global warming level simulations, as they follow the SSP1-2.6 scenario for both warming levels.

Table 1. Simulated $CO_2$-fe emissions in 2025, 2030, 2050 and averaged between 2100-2150 in the 1.5 °C and 2.0 °C warming level simulations. The last column indicates the minimum of the derivative of the $CO_2$-fe curve, between 2026 and 2100. For models that performed several ensemble members, the ensemble mean is indicated, as well as the minimum to maximum range in the brackets.

| Models | $CO_2$-fe (PgC yr$^{-1}$) 2025 | $CO_2$-fe (PgC yr$^{-1}$) 2030 | | $CO_2$-fe (PgC yr$^{-1}$) 2050 | | $CO_2$-fe (PgC yr$^{-1}$) 2100-2150 | | max. $CO_2$-fe reduction (PgC yr$^{-2}$) 2026-2100 | |
|---|---|---|---|---|---|---|---|---|---|
| | | 1.5 °C | 2.0 °C | 1.5 °C | 2.0 °C | 1.5 °C | 2.0 °C | 1.5 °C | 2.0 °C |
| CESM2 | 13.2 | 9.2 | 12.9 | -2.0 | 6.5 | -0.5 | -1.5 | -3.2 | -0.3 |
| EC-Earth ensemble | 13.1 | 7.2 | 11.8 | -3.6 | 3.8 | 0.8 | 0.7 | -2.0 | -0.8 |
| | | (4.3 to 8.8) | (11.8 to 11.8) | | | | | | (-0.8 to -0.8) |
| NASA-GISS | 18.1 | 11.9 | 17.1 | -2.3 | 1.3 | 0.8 | 2.3 | -3.6 | -2.3 |
| GFDL-ESM2M ensemble | 11.8 | 10.9 | 12.0 | 5.0 | 10.8 | 0.6 | 2.2 | -0.4 | -0.5 |
| | | (0.8 to 11.2) | (11.9 to 12.0) | (3.5 to 7.3) | (10.3 to 11.0) | (-0.8 to 2.6) | (1.2 to 3.8) | (-0.6 to -0.2) | (-1.3 to -0.1) |
| GFDL-ESM4 | 11.8 | 9.3 | 11.8 | 4.2 | 9.6 | 1.4 | 2.7 | -0.7 | -0.1 |
| IPSL ensemble | 11.0 | 6.7 | 10.9 | 1.4 | 8.0 | 2.9 | 2.1 | -0.9 | -0.3 |
| | (9.6 to 12.4) | (5.7 to 7.7) | (9.4 to 12.4) | (0.6 to 2.3) | (7.8 to 8.3) | (2.8 to 3.0) | (1.9 to 2.2) | (-1.0 to -0.8) | (-0.3 to -0.3) |
| MIROC | 12.2 | 8.4 | 13.1 | -2.8 | 6.4 | 2.9 | 2.7 | -3.5 | -1.0 |
| MPI | 13.7 | 9.6 | 13.5 | 1.9 | 9.5 | 2.7 | 2.6 | -1.4 | -0.2 |
| NorESM2 | 13.2 | 3.1 | 13.0 | -2.6 | 7.0 | -1.7 | 2.2 | -2.2 | -0.5 |
| Bern ensemble | 12.9 | 7.3 | 12.6 | -0.7 | 6.6 | 0.8 | 0.8 | -1.7 | -0.6 |
| | (0.7 to 11.5) | (12.0 to 12.9) | (-4.4 to 5.3) | (-0.9 to 10.6) | (-0.7 to 2.5) | (-1.0 to 2.8) | (-3.2 to -0.7) | (-1.9 to -0.1) | |
| UVic | 12.6 | 7.7 | 10.9 | 1.9 | 11.4 | 1.4 | 1.5 | -1.1 | -0.6 |
| HadCM3-FaIR2 ensemble | 18.9 | 6.1 | 16.2 | 1.5 | 5.9 | 0.3 | 1.7 | -3.5 | -0.7 |
| | (12.1 to 27.4) | (-3.6 to 13.4) | (10.4 to 25.2) | (-6.0 to 7.1) | (-2.6 to 9.8) | (-1.8 to 2.0) | (-2.2 to 4.9) | (-6.2 to -1.7) | (-2.2 to -0.3) |
| Multi-model mean | 13.7 | 8.1 | 13.0 | 0.2 | 7.2 | 1.0 | 1.7 | -2.0 | -0.7 |
| Multi-model range | (11.0 to 18.9) | (3.1 to 11.9) | (10.9 to 17.1) | (-3.6 to 5.0) | (1.3 to 11.4) | (-1.7 to 2.9) | (-1.5 to 2.7) | (-3.6 to -0.4) | (-2.3 to -0.1) |

The simulated model-specific $CO_2$ emissions from land use change and $CO_2$-fe emissions from non-$CO_2$ agents are not available as output from all models. Nevertheless, negative $CO_2$-fe emissions from non-$CO_2$ radiative forcing agents are only possible if the radiative forcing from these agents is decreasing. However, the radiative forcing agents follow the SSP1-2.6 or RCP2.6 scenario and after 2100 they are set to constant values for all models suggesting that substantial negative non-$CO_2$ radiative forcing is unlikely post-2100. In addition, land use area is also set to be constant after 2100. Considering that $CO_2$ emissions from both land use change and non-$CO_2$ agents are likely to be zero or slightly above zero after temperature stabilization, it is reasonable to conclude that the positive $CO_2$-fe and fossil fuel $CO_2$ emissions after stabilizing warming are more likely a consequence of the overall negative zero emissions commitment rather than net negative forcing from non-$CO_2$ forcing or from land use change.

The uncertainty across models found here in residual fossil fuel emissions (Fig. 4a,b) compatible with temperature stabilization appears to be in agreement with the results from Jenkins et al. (2022b), who diagnosed compatible emissions with halting warming after 1pctCO2 and 1.5ºC-compatible $CO_2$ emission-driven experiments. They calculate these emissions based on a theoretical framework (defining the RAZE parameter - rate of adjustment to zero emissions), results from the ZECMIP simulations, and a climate emulator. In their scenario in which non-$CO_2$ forcing agents follow SSP1-1.9 and $CO_2$ emissions linearly decrease to zero between 2021 and 2050, the best estimate of $CO_2$ emissions compatible with halting warming after 2050 is given at 0.66 PgC yr$^{-1}$ (2.2 GtCO$_2$ yr$^{-1}$), with a 5th-95th percentile range of -2.2 to 1.9 PgC yr$^{-1}$. In their formulation of the multi-decade ZEC response, this equates to a small negative RAZE parameter whose uncertainty spans zero. On the other hand, Mengis et al. (2018) find that net negative fossil fuel $CO_2$ emissions are necessary to stabilize global surface temperature at 1.5 ºC in an observation-constrained carbon cycle perturbed-parameter ensemble of the UVic model. In their study, they used RCP2.6 non-$CO_2$ forcing extended to the year 2200, which in total causes positive radiative forcing between the first time the temperature is reached in 2055, and the period of stabilisation until 2200. They discuss that this is a likely cause for the net-negative $CO_2$ emissions.

### 3.4 Consequences for atmospheric $CO_2$

Since the simulations are $CO_2$ emission-driven, atmospheric $CO_2$ evolves dynamically (Fig. 4e,f). The multi-model mean atmospheric $CO_2$ reaches 420 ppm in 2020, slightly higher than the observed 412 ppm (Lan et al., 2023). In both temperature stabilization simulations, the multi-model mean exhibits a peak and subsequent decline behavior. Atmospheric $CO_2$ peaks at 438 ppm in 2031 for the 1.5 ºC scenario and at 499 ppm in 2070 for the 2.0 ºC scenario, subsequently decreasing to 410 ppm in the 1.5 ºC scenario and to 480 ppm in the 2.0 ºC scenario by 2150. Thus, atmospheric $CO_2$ should start to decrease around 2030 to reach the 1.5 ºC level, according to the multi-model mean. Some models do not simulate such a smooth peak and decline behavior, and simulate in addition a (temporary) rise in atmospheric $CO_2$ due to a temporary rise in $CO_2$ emissions during or after the decline (Fig. 4a).

The different $CO_2$ emissions and the strength of the land and ocean carbon sinks lead to large differences in atmospheric $CO_2$. The model min-max range in atmospheric $CO_2$ of 53 ppm in year 2025 originates from different ocean and land carbon sinks and $E_{LUC}$ emissions during the historical period (Hoffman et al., 2014) as all models have identical prescribed fossil fuel

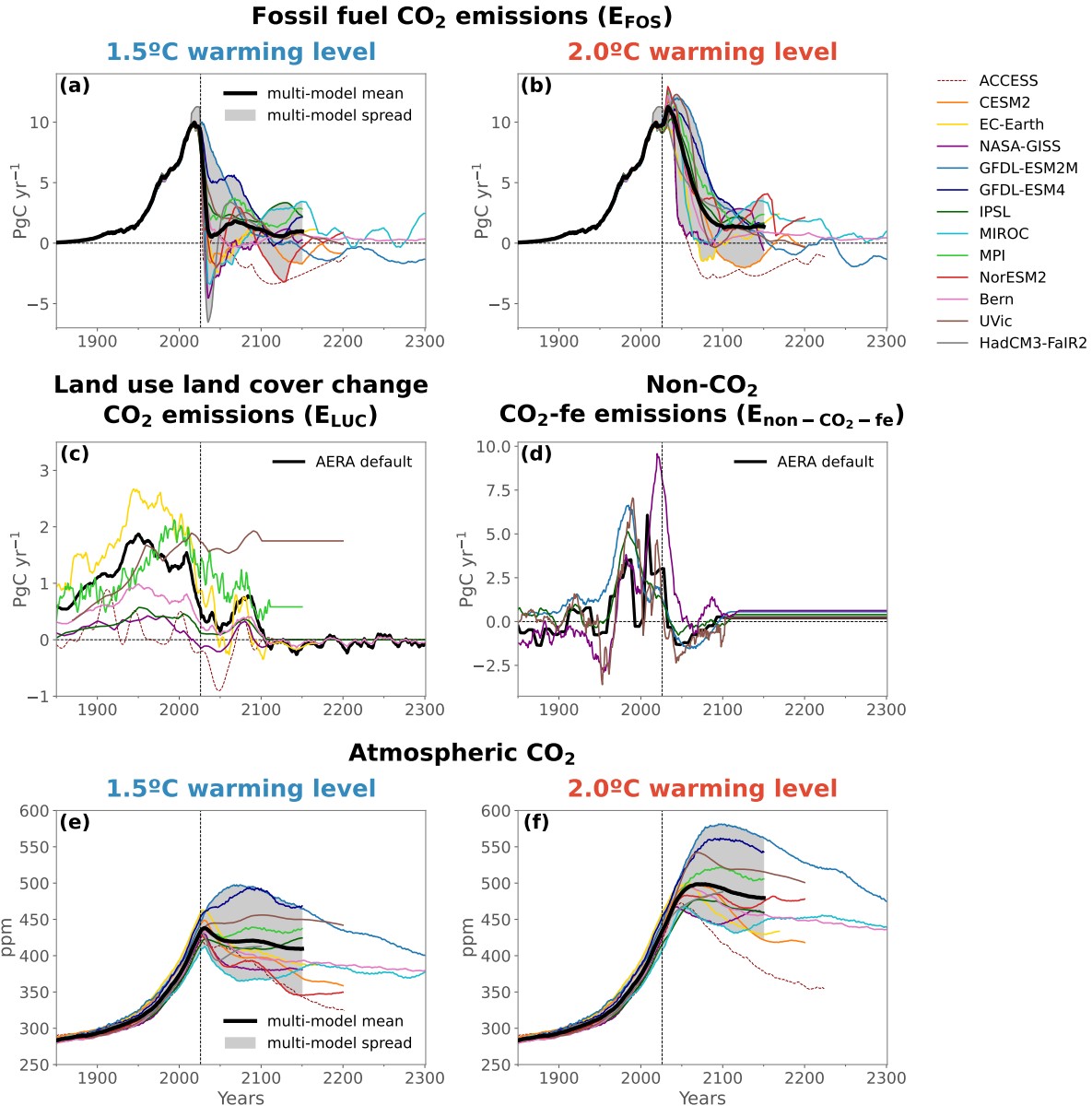

**Figure 4.** Simulated fossil fuel $CO_2$ emissions, emissions from land use change and non-$CO_2$ forcing agents, and atmospheric $CO_2$. The multi-model mean, excluding the ACCESS model, is displayed by the black thick line, with the grey shading indicating the min-max spread. The ensemble mean is shown for models that have several ensemble members. The vertical dotted line at the year 2026 marks the beginning of the AERA simulations. In panels c and d, the black line shows the default AERA input for both warming levels, whereas the colored lines show the diagnosed land-use change or non-$CO_2$ forcing equivalent emissions for models that do not use AERA default values. The (c) non-$CO_2$ and the (d) land-use change $CO_2$-fe emissions follow identical trajectories for both warming levels.

$CO_2$ emissions (Fig. 4a,b; note HadCM3-FaIR2 differs due to the emissions being diagnosed and not prescribed, see Appendix A). After year 2025, the spread in atmospheric $CO_2$ is also driven by the divergent $E_{FOS}$ pathways and thus strongly increases, as $E_{FOS}$ evolves also with the model-dependent AERA calculation every five years. The model min-max range expands to 105 ppm (1.5 ℃ simulation) and 55 ppm (2.0 ℃) by year 2050, 125 ppm (1.5 ℃) and 139 ppm (2.0 ℃) by year 2100, and 123 ppm (1.5 ℃) and 141 ppm (2.0 ℃) by year 2150.

Part of these large uncertainties may stem from internal variability (Fig. 3c and B3c). GFDL-ESM2M's 5 ensemble members show a min-max range of 54 ppm (1.5 ℃) and 49 ppm (2.0 ℃) by 2100, approximately 43 % and 40 %, respectively, of the multi-model range. EC-Earth's 3 ensemble members represent 29 % and 27 % of the model range in 2100. GFDL-ESM2M's range tends to decrease over the 23$^{rd}$ century, reaching 24 ppm (1.5 ℃) and 41 ppm (2.0 ℃) by 2300. Sensitivity simulations with the Bern model, covering the IPCC AR6 likely range of TCRE, exhibit an inter-member spread in atmospheric $CO_2$ levels that is as large as the inter-model spread, persisting during stabilization (not shown). Thus, while internal variability certainly plays an important role by 2100, differences in TCRE among the models remain the primary driver for the substantial model uncertainties in simulated atmospheric $CO_2$. Additional processes may be at play in explaining the model spread, not explained by differences in TCRE, such as the sensitivity to evolving non-$CO_2$ agents and the representation of land use land cover changes in the models. For example, the MIROC model has a low TCRE and negative ZEC (MacDougall et al., 2020), and yet it exhibits the lowest levels of atmospheric $CO_2$ at the end of the historical period (Fig. 4e,f).

## 4   The remaining emission budget from 2020 to 2150

The AERA simulations provide actual emissions data both during the period of model stabilization to the prescribed warming level and beyond. We compare these numbers with the remaining carbon budget (RCB) estimates from the IPCC AR6 Working Group I (WGI; IPCC (2021)). For consistency with IPCC data, we convert our results from GtC into $GtCO_2$, and round up the numbers to the nearest 50 $GtCO_2$. For the 1.5 °C warming level, our median estimate of cumulative total $CO_2$-fe emissions between 2020 and 2150 is 900 $GtCO_2$, with a 17$^{th}$-83$^{rd}$ percentile model range of 450 to 1800 $GtCO_2$ (Table 4). This includes 700 (250 to 1450) $GtCO_2$ from fossil fuel $CO_2$, 150 (50 to 300) $GtCO_2$ from land use changes, and 10 (-10 to 20) $GtCO_2$ from non-$CO_2$ forcing agents. Note the sum of the individual median estimates are not exactly equal to the total $CO_2$-fe emissions because the median is not linear, and because of rounding. The $CO_2$ only budget from both fossil fuel and land use change emissions is 800 (250 to 1800) $GtCO_2$. In comparison, the IPCC AR6 WGI estimate of the remaining carbon budget ($CO_2$ only) from year 2020 is 500 (300 to 900) $GtCO_2$ for a 50 % (83 to 17 %) likelihood of limiting global warming to 1.5 °C. The methodological update from Forster et al. (2024) slightly lowers the budget to 400 (200 to 750) $GtCO_2$ (Table 2). Our new median remaining carbon budget ($CO_2$ only) estimate from year 2020 for the 1.5 °C warming level is therefore 60 % larger than the median IPCC AR6 estimate.

For the 2.0 °C warming level, the AERA simulations estimate median (17$^{th}$-83$^{rd}$ percentile range) total $CO_2$-fe emissions between 2020 and 2150 at 2150 (1600 to 2900) $GtCO_2$. 2050 (1350 to 2600) $GtCO_2$ is from fossil fuels, 150 (50 to 300) $GtCO_2$ from land use changes, and 10 (-10 to 20) $GtCO_2$ from non-$CO_2$ forcing agents (note the land use change and non-$CO_2$

emissions are not warming level dependant in the AERA protocol). This gives a total $CO_2$-only budget of 2250 (1350 to 2900) $GtCO_2$, compared to 1350 (900 to 2350) $GtCO_2$ from the IPCC AR6 RCB, and 1300 (900 to 2200) from Forster et al. (2024). Our new median remaining carbon budget ($CO_2$ only) estimate from year 2020 for the 2.0 °C warming level is therefore 67 % larger than the median IPCC AR6 estimate.

**Table 2.** Estimates of the remaining emission budget for 1.5 °C and 2.0 °C warming levels, considering the AERA-MIP simulations from the beginning of year 2020 until the end of year 2150. Reported here is the multi-model median, as well as the 17[th]-83[rd] percentile model range in the brackets, for each term. The ACCESS and HadCM3 models are excluded from the model statistics (ACCESS simulations do not converge to the given warming level in time, see text; HadCM3 simulations stop in 2100). In this table, we use the default time series prescribed to AERA for $E_{LUC}$ and for $E_{non\text{-}CO_2\text{-}fe}$ for the models which do not have their own internal estimate (see Methods and Appendix). For a few models, we use the corrected $E_{LUC}$ time series (see Appendix A). All values are given here in $GtCO_2$ (a factor 3.67 compared to PgC), and rounded to the nearest 50 $GtCO_2$ (except for $E_{non\text{-}CO_2\text{-}fe}$ since the range is smaller than 50 $GtCO_2$), for direct comparison with IPCC AR6 WGI estimates of the remaining carbon budget, as well as the update from Forster et al. (2024). For those two references, the reported numbers correspond to a 50 % (83 % to 17 %) likelihood of limiting global warming to the temperature limit.

| | Warming level | Total $CO_2$-fe $E_{FOS} + E_{LUC} + E_{non\text{-}CO_2\text{-}fe}$ | FF+LUC $CO_2$ $E_{FOS} + E_{LUC}$ | FF $CO_2$ $E_{FOS}$ | LUC $CO_2$ $E_{LUC}$ | non-$CO_2$ $E_{non\text{-}CO_2\text{-}fe}$ |
|---|---|---|---|---|---|---|
| This study | 1.5 °C | 900 | 800 | 700 | 150 | 10 |
| | | (450 to 1800) | (250 to 1800) | (250 to 1450) | (50 to 300) | (-10 to 20) |
| IPCC AR6 WGI | 1.5 °C | | 500 | | | 220-440 |
| | | | (300 to 900) | | | |
| Forster et al. (2024) | 1.5 °C | | 400 | | | |
| | | | (200 to 750) | | | |
| This study | 2.0 °C | 2150 | 2250 | 2050 | 150 | 10 |
| | | (1600 to 2900) | (1350 to 2900) | (1350 to 2600) | (50 to 300) | (-10 to 20) |
| IPCC AR6 WGI | 2.0 °C | | 1350 | | | 220-440 |
| | | | (900 to 2350) | | | |
| Forster et al. (2024) | 2.0 °C | | 1300 | | | |
| | | | (900 to 2200) | | | |

405    The AERA simulations show larger $CO_2$ emission budgets for both 1.5 °C and 2.0 °C warming levels compared to the IPCC AR6 RCBs, although the large model range encompasses the uncertainty distribution from AR6. The methodologies are different and part of that gap could be due to the choice of non-$CO_2$ scenario. In the AERA simulations, the remaining total $CO_2$-fe emission budget calculated in step 2 of the AERA is independent of the future pathway of non-$CO_2$ forcing agents. Nonetheless, the partition between future $CO_2$ and non-$CO_2$ emissions is scenario-dependent, with a very small amount

410 distributed to non-$CO_2$ emissions here (median estimate of 10 $GtCO_2$), constrained by the SSP1-2.6 scenario in our modelling protocol. Similarly, the AR6 estimate is also scenario-dependent regarding future additional non-$CO_2$ warming until net-zero $CO_2$ emissions. The median estimate for this non-$CO_2$ warming is 0.1-0.2 ℃ (Canadell et al., 2021), corresponding to 220-

GtCO$_2$ following the median TCRE, much more than our median estimate of 10 GtCO$_2$. The scenario uncertainty in the AR6 estimate represents variations of at least $\pm$220 GtCO$_2$. Thus, when adding up CO$_2$ and non-CO$_2$ contributions (although estimated using different approaches), the median 1.5 ℃ remaining emission budget is in fact very similar between the AR6 estimate and the AERA simulations (around 800 GtCO$_2$). This does not hold for the 2.0 ℃ remaining emission budget, which is still larger in the AERA simulations, even when accounting for the differences in non-CO$_2$ emission budget.

Another source of difference can stem from the uncertainty in the response to CO$_2$ emissions. Interestingly, the median TCRE (CO$_2$-only) from the models participating in AERA and sampled in Table 2 (1.60 ℃ / 1000 PgC) is very close to the IPCC AR6 median TCRE estimate (1.65 ℃ / 1000 PgC), indicating that model sampling bias alone cannot explain the differences. In addition, the RCB calculation in AR6 relies on the assumption that TCRE remains constant over time. The numbers from the AERA simulations provided in Table 2 on the other hand retrospectively count the simulated emissions until temperature is stabilized and beyond. This makes AERA a valuable tool for exploring how TCRE and the estimated emission budget evolve as the climate responds to mitigated emissions and temperature stabilization. Next, we explore the evolution of the remaining CO$_2$-fe emission budget.

## 5   The evolving remaining emission budget

AERA calculates the simulated remaining CO$_2$-fe emission budget (REB) every five years based on the remaining allowable warming and the TCRE-fe relationship (see Equation 2 in Methods). At the beginning of year 2026, the start of the AERA period, the multi-model mean REB is 99 (47 to 204) PgC for the 1.5 ℃ warming level and 409 (248 to 581) PgC for the 2.0 ℃ level (Table 3). These REBs translate to 7 (3 to 14) years of CO$_2$-fe emissions sustained at mean 2020 levels (14.1 PgC yr$^{-1}$) for the 1.5 ℃ warming level and 29 (18 to 41) years for the 2.0 ℃ level before reaching the respective temperature level.

To compare these model-based estimates to other REB estimates from the beginning of year 2021 (Jenkins et al., 2021; Terhaar et al., 2022a), we combine the CO$_2$-fe cumulative emissions of 70 PgC on average from 2021 to 2025 with the REB from beginning of year 2026, which yields a mean REB of 169 (113 to 264) PgC from the start of 2021 for the 1.5 ℃ warming level and 479 (309 to 640) PgC for the 2.0 ℃ level. Our model-estimated REB range encompasses the AERA-based REB calculations from observations of 167 PgC for 1.5 ℃ and 472 PgC for 2.0 ℃ (Terhaar et al., 2022a). The model's estimated REB also aligns with observation-constrained estimates by Jenkins et al. (2021) of 128-237 PgC for an 83 %-17 % probability of limiting warming to 1.5 ℃ (also based on total CO$_2$-fe emissions).

In our simulations, the multi-model mean REB as estimated at the beginning of year 2026 ($EB_{2026}$) is smaller than the emissions that are actually emitted until temperature stabilization. Or in other words, the actual EB is larger than the REB estimated at the beginning of 2026 (Table 3). Between 2100 and 2150, the multi-model mean estimate of $EB_{2026}$ (calculated as the sum of the REB in the respective year and the already emitted emissions between 2026 and that year, see Methods for details) reaches 217 (-40 to 475) PgC for 1.5 ℃ and 552 (297 to 778) PgC for 2.0 ℃. These updated emission budgets correspond to 15 years (-3 to 34) of sustained mean CO$_2$-fe emissions at 2020 levels for 1.5 ℃ and 39 years (21 to 55) for 2.0 ℃. Thus, the simulated remaining emissions until temperature stabilization are 2.2 times larger than estimated at the beginning

**Table 3.** Remaining $CO_2$-fe Emission Budget (REB) from beginning of year 2026, calculated at the first stocktake (end of year 2025; $EB_{2026}(2025)$), and recalculated and averaged between 2100 and 2150, taking into account the emissions actually seen by the models since 2026 ($EB_{2026}(2100\text{-}2150)$; see Eq. 4).

| Models | REB at first stocktake $EB_{2026}(2025)$ (PgC) | | $EB_{2026}$ averaged between stocktakes 2100-2150 $EB_{2026}(2100\text{-}2150)$ (PgC) | |
|---|---|---|---|---|
| | 1.5 °C | 2.0 °C | 1.5 °C | 2.0 °C |
| CESM2 | 100 | 345 | 6 | 298 |
| EC-Earth ensemble | 81 | 339 | 78 | 310 |
| | (52 to 97) | (322 to 359) | (16 to 142) | (230 to 365) |
| NASA-GISS | 132 | 470 | 112 | 444 |
| GFDL-ESM2M ensemble | 204 | 581 | 353 | 778 |
| | (178 to 244) | (534 to 615) | (260 to 423) | (678 to 889) |
| GFDL-ESM4 | 111 | 407 | 475 | 756 |
| IPSL ensemble | 60 | 248 | 412 | 585 |
| | (58 to 62) | (240 to 256) | (386 to 438) | (561 to 609) |
| MIROC | 86 | 478 | 259 | 599 |
| MPI | 104 | 416 | 416 | 721 |
| NorESM2 | 47 | 442 | -40 | 677 |
| Bern ensemble | 86 | 374 | 118 | 418 |
| | (24 to 174) | (236 to 554) | (-133 to 396) | (109 to 792) |
| UVic-ESCM | 120 | 439 | 284 | 549 |
| HadCM3-FaIR2 ensemble | 58 | 369 | 133 | 491 |
| | (-12 to 148) | (230 to 606) | (-89 to 276) | (242 to 718) |
| Multi-model mean | 99 | 409 | 217 | 552 |
| Multi-model range | (47 to 20) | (248 to 581) | (-40 to 475) | (297 to 778) |

of 2026 for the 1.5 ℃ warming level and 1.4 times larger for 2.0 ℃. This increase in the $EB_{2026}$ between the beginning of the AERA period and the end of the simulations corresponds to a decrease in the TCRE-fe (Fig. B4), which is qualitatively consistent with a slightly negative multi-decade zero emissions commitment (and RAZE parameter) (MacDougall et al., 2020; Jenkins et al., 2022b) and the resulting residual positive emissions found on average across models during the stabilization

phase (Fig. 2). However, this relationship does not hold for all models, pointing to other processes that may be at play, such as the evolution of physical feedbacks, heat uptake by the ocean, carbon uptake by the ocean and land sinks, and the fraction of radiative forcing explained by $CO_2$ compared to non-$CO_2$ agents (e.g. Williams et al., 2017, 2020).

When summed from 1850 to 2150, the total $CO_2$-fe emissions amount to 1063 (750 to 1461) PgC for the 1.5 ℃ warming level and 1380 (1087 to 1785) PgC for the 2.0 ℃ level. The lower end of the multi-model distribution encompasses the estimate

of 817 PgC, respectively 1090 PgC found by Mengis and Matthews (2020) under a 1.5 ℃ and 2.0 ℃ stabilisation scenario with the UVic model.

Similar to residual emissions compatible with temperature stabilization, the spread across models in $EB_{2026}$ is very large, which does not only reflect model differences but also uncertainties due to internal climate variability. The range in $EB_{2026}$, when estimated at the end of year 2025, across the ensemble members of GFDL-ESM2M, represents 41 % (24 %) of the total

model range for the 1.5 ℃ (2.0 ℃) warming level (Fig. B5). For EC-Earth, the ensemble range amounts to 28 % (11 %) of the total model range. These differences in $EB_{2026}$ are predominantly caused by differences in estimated anthropogenic warming in 2020 due to the difference in internal variability in each ensemble member. If a perfect fit to GSAT for anthropogenic warming existed, which could remove all internal variability, these differences would vanish. This shows how sensitive the emission budget is to the estimate of global warming (Tokarska et al., 2020). When $EB_{2026}$ is re-estimated later in the simulations and

averaged between 2100 and 2150, the GFDL-ESM2M range is 41 % (44 %) of the model range and the EC-Earth ensemble amounts to 24 % (31 %).

The model spread in the change of $EB_{2026}$ over time is also large. For the 1.5 °C warming level, the model range spans from a $EB_{2026}$ decrease by a factor of -1.2 to an increase of a factor of 6.9 between 2025 and 2100-2150. The largest decrease in absolute value is simulated by CESM2, for which $EB_{2026}$ is initially estimated at 100 PgC, but decreases drastically to 6 PgC

by 2100-2150 for the 1.5 ℃ warming level. The largest increase in absolute value is simulated by GFDL-ESM4, for which $EB_{2026}$ is initially 111 PgC but increases sharply to 475 PgC. For the 2 °C warming level, the factor spans from 0.8 to 2.4. These differences in the evolution of the emission budget reflect uncertainties in the non-linear evolution of the TCRE-fe, in the response to non-$CO_2$ forcing agents but also partly in internal climate variability. Within the GFDL-ESM2M 5-member ensemble, the increase in $EB_{2026}$ ranges from a factor 1.2 to 2.4 for the 1.5 ℃ warming level, and from a factor 1.1 to 1.7 for

the 2 ℃ warming level.

As the evolution in the remaining emission budget is crucial for achieving goals defined in the Paris Agreement, we here test if this evolution can be explained by standard climate metrics, such as the (effective) Equilibrium Climate Sensitivity (ECS), the Transient Climate Response (TCR) and the ($CO_2$-only) Transient Climate Response to cumulative $CO_2$ emissions (TCRE). These metrics, reported by various studies (Arora et al., 2020; Meehl et al., 2020; MacDougall et al., 2020) are

analyzed alongside the 2025 estimate of the TCRE-fe including all $CO_2$-fe emissions. For the 2.0 ℃ warming level, there

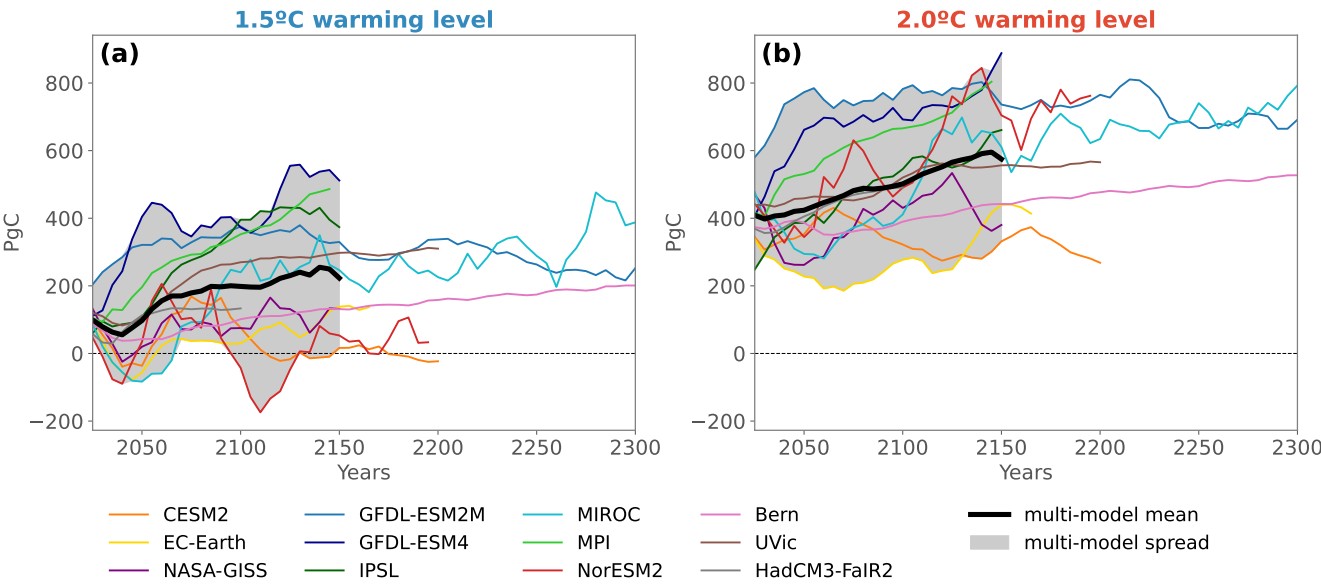

**Figure 5.** Remaining $CO_2$-fe Budget recalculated from the beginning of year 2026 at each stocktake (denoted as $EB_{2026}$ in the text) in the 1.5 ºC (a) and 2.0 ºC (b) warming level simulations. The ensemble mean is shown for models that have several ensemble members. The multi-model mean is displayed by the black thick line and the grey shading covers the min-max spread.

is no strong relationship between those metrics and the differences in $EB_{2026}$ between the 2100-2150 average and the first estimate at the end of 2025 (Fig. 6). The IPSL model stands out as being a highly sensitive model (Boucher et al., 2020), thus with a very low initial estimate of the $EB_{2026}$ compared to other models. However, as emissions rapidly decrease during the AERA simulations, the linearity of the TCRE-fe relationship breaks in this model and positive emissions become necessary

to keep warming the model at the prescribed warming level. This could be due either to large negative ZEC (which needs to be confirmed by dedicated ZECMIP simulations), and/or to strong sensitivity to non-$CO_2$ greenhouse gases which are heavily mitigated in the SSP1-2.6 scenario. When the IPSL model is removed from the analysis, a clearer relationship emerges. Models with higher sensitivity (e.g., high ECS/TCR/TCRE/TCRE-fe) tend to exhibit a lower increase or even a slight decrease in the $EB_{2026}$, while less sensitive models show a more substantial increase. It is noticeable that the relationship between the different

configurations of the Bern model (small pink dots) aligns particularly well with the slope of the linear regression between models excluding IPSL (red line). This relationship aligns with previous findings illustrating a positive correlation between the zero emissions commitment (implying decreases in EB over time) and TCR or TCRE across multiple models (MacDougall et al., 2020). Additionally, this correlation is evident between the zero emissions commitment and ECS within parameter-perturbed ensembles of a single model (MacDougall et al., 2020), as also shown here for the Bern model. Nevertheless, this

relationship is not true for all models as shown in Fig. 6 and the range remains large. For example, NASA-GISS has a low

sensitivity but a decreasing $EB_{2026}$. As pointed out by MacDougall et al. (2020), the evolution of the surface temperature and thus the evolving amount of emissions needed to stabilize warming is a balance between large quantities controlled by heat and carbon dynamics, and thus not expected to scale particularly well with these climate metrics.

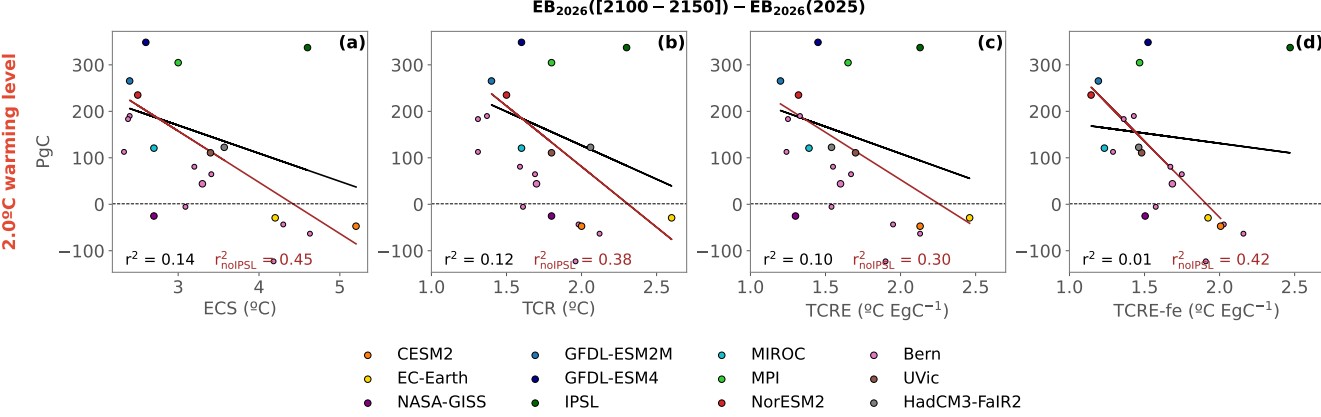

**Figure 6.** Simulated changes in the $EB_{2026}$, i.e., the estimate averaged over 2100-2150 minus the first estimate at the end of year 2025, plotted against ECS (a), TCR (b), $CO_2$-only TCRE (c) and TCRE-fe at the end of year 2025 (d). Here only the 2.0 °C simulation is shown. Results for the 1.5 °C simulation are displayed in Fig. B6. The black line indicates the linear regression including all models, and the red line excluding the IPSL model. The 9 parameter-perturbed configurations of the Bern model are displayed by the small pink dots but the regression is calculated based on the ensemble mean (larger pink dot).

## 6    Where does the carbon go?

### 6.1    Global response

Atmosphere, ocean and land carbon fluxes exhibit peak and decline patterns, albeit with distinct differences between all three sinks (Fig. 7). In this section, our emphasis is on the 2.0 °C scenario. In addition, the analysis for the 1.5 °C scenario is shown in the Appendix (Fig. B7), and the numbers for the cumulative emissions and sinks are reported for both scenarios in Tables 4 and 5.

The atmospheric $CO_2$ growth rate, $G_{ATM}$, initially mirrors fossil fuel emissions, reaching 4.7 (3.5 to 5.7) PgC yr$^{-1}$ in 2020 (Fig. 7a), consistent with the observation-based estimate of $5.0 \pm 0.2$ PgC yr$^{-1}$ (Friedlingstein et al., 2022). As emissions decrease, $G_{ATM}$ deviates from fossil fuel emission trends, turning negative and reaching -0.5 (range: -2.7 to 1.7) PgC yr$^{-1}$ between 2100-2150 due to continued carbon uptake by the ocean and land. The ocean remains a consistent carbon sink, with multi-model mean uptake decreasing from 2.9 (2.2 to 3.3) PgC yr$^{-1}$ in 2020 to 1.2 (0.2 to 1.9) PgC yr$^{-1}$ by 2100-2150 (Figs. 7c). Conversely, the net land carbon sink ($S_{LAND}$-$E_{LUC}$) undergoes a transition from a net source to a sink between 1850-1950. The net land carbon sink peaks at 2.1 (0.5 to 4.1) PgC yr$^{-1}$ in 2020 and remains a net sink or neutral until 2200 (Fig. 7g).

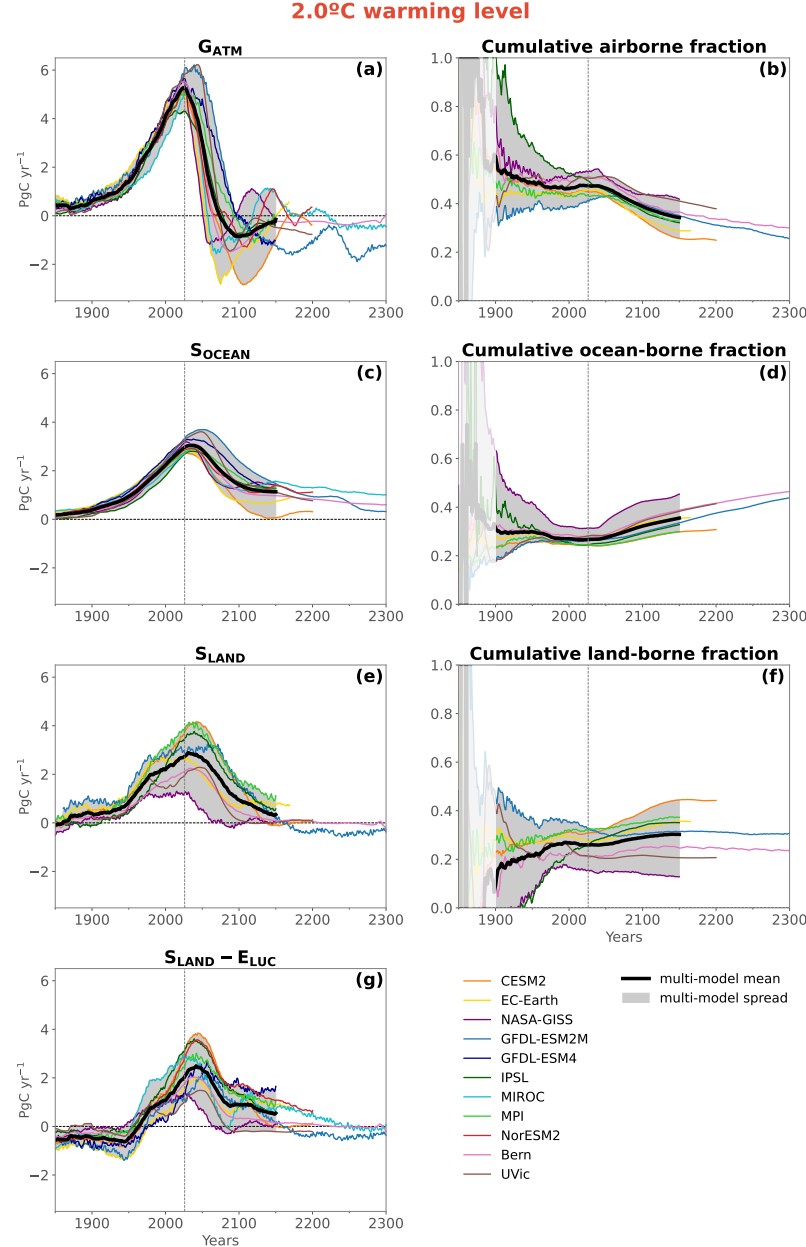

**Figure 7.** (a) Atmospheric $CO_2$ growth rate ($G_{ATM}$), (c) net ocean sink ($CO_2$ flux into the ocean; $S_{OCEAN}$), (e) gross land sink ($S_{LAND}$), (g) net land sink ($CO_2$ flux into land; $S_{LAND} - E_{LUC}$), cumulative (b) airborne, (d) ocean-borne and (f) land-borne fractions, in the 2.0 ºC simulation. The multi-model mean is displayed by the black thick line and the grey shading covers the min-max spread. Apart from the cumulative fractions, all time series have been smoothed with a 31-year running mean to remove short-term internal variations for visual purposes. The numbers are reported in the text without the 31-year average to be comparable with the Global Carbon Budget estimates.

Notably, the land sink reaches a neutral state faster than the ocean sink, even temporarily shifting to a small carbon source by the end of the 22$^{nd}$ century. The simulated ocean and land carbon uptake in 2020 align well with the observation-based estimates: 3.0 $\pm$ 0.4 PgC for ocean carbon uptake and 2.0 PgC yr$^{-1}$ ($S_{LAND}$ = 2.9 $\pm$ 1.0 PgC yr$^{-1}$ minus $E_{LUC}$ = 0.9 $\pm$ 0.7 PgC yr$^{-1}$) for land carbon uptake (Friedlingstein et al., 2022). Among the models providing land use change emissions estimates, the gross land carbon uptake $S_{LAND}$ emerges as a larger sink than the net land carbon flux $S_{LAND} - E_{LUC}$ (Fig. 7e), trending towards neutrality or even displaying a source tendency on longer timescales in GFDL-ESM2M. $S_{LAND}$ in NASA-GISS, which lies at the lower end of the model range, is small because its sensitivity to $CO_2$ fertilization is damped: while the model simulates enhanced photosynthetic uptake of $CO_2$, its vegetation structure remains fixed and the extra carbon is instead allocated to the soil where it can still be respired back to the atmosphere. Also, the NASA-GISS model does not capture regrowth from secondary forest. These together lead to a relatively smaller cumulative land-borne fraction (Fig. 7f) and larger cumulative ocean-borne fraction by compensation (Fig. 7d).

The net ocean carbon flux initially has a narrow spread across models during the historical period, with a min-max range of 0.8 PgC yr$^{-1}$ over 2000-2020. However, this spread notably widens after 2025, reaching 1.5 PgC yr$^{-1}$ over 2040-2060 and 1.7 PgC yr$^{-1}$ over 2100-2150. Pre-2025 differences among models are mainly due to various representations of ocean circulation and biogeochemistry in ESMs (Terhaar et al., 2022b). Post-2025, the widening spread results also from differing emission pathways. In contrast, the land carbon flux exhibits a much wider spread during the historical period, reaching 4.0 PgC yr$^{-1}$ over 2000-2020. This range only increases slightly after 2025 to 5.0 PgC yr$^{-1}$ over 2040-2060, reverting to 4.0 PgC yr$^{-1}$ over 2100-2150. The substantial spread in land carbon flux, not entirely depicted in Fig. 7 due to a 31-year average for visual clarity, arises from the diverse representations of land carbon processes across models (Canadell et al., 2021).

From 1850 to 2020, 294 PgC (ranging from 236 to 354 PgC) of the total cumulative $CO_2$ emissions from fossil fuels (462 PgC) and land use change (170 PgC) remained in the atmosphere. During the same period, the ocean has taken up 173 PgC (ranging from 128 to 208 PgC), and the land absorbed 168 PgC (ranging from 79 to 254 PgC) (Table 5). This partitioning translates to a cumulative airborne fraction of 0.48 (0.41 to 0.53), an ocean-borne fraction of 0.27 (0.24 to 0.31) and a land-borne fraction of 0.26 (0.16 to 0.33). These values are similar to observation-based estimates for 1850-2020 of 0.26 for the ocean sink and 0.30 for the land sink (Friedlingstein et al., 2022). Over the simulation period, the cumulative airborne fraction steadily decreases alongside declining emissions after 2025, signifying the gradual uptake of anthropogenic carbon by the ocean and the land (Fig. 7b,d,f). By the year 2150, the cumulative airborne fraction decreases to 0.34 (0.26 to 0.42), with an average of 416 (284 to 354) PgC of anthropogenic carbon still remaining in the atmosphere. Concurrently, the ocean-borne fraction consistently rises, emerging as the dominant fraction by the end of the simulations. In 2150, the ocean-borne fraction increases to 0.36 (0.30 to 0.45), surpassing both the land-borne fraction of 0.30 (0.13 to 0.44) and the airborne fraction. The cumulative ocean sink by 2150 amounts to 426 (330 to 556) PgC, while the land has taken up 370 (113 to 540) PgC. Nonetheless, substantial variability persists among model estimates for each fraction.

**Table 4.** Cumulative $E_{FOS}$ and $E_{LUC}$ emissions in 2020 and 2150.

| Models | $E_{FOS}$ (PgC) | | | $E_{LUC}$ (PgC) | | $E_{FOS} + E_{LUC}$ (PgC) | | |
|---|---|---|---|---|---|---|---|---|
| | 2020 | 2150 | | 2020 | 2150 | 2020 | 2150 | |
| | | 1.5°C | 2.0°C | | | | 1.5°C | 2.0°C |
| CESM2 | 460 | 522 | 800 | 210 | 242 | 670 | 764 | 1042 |
| EC-Earth ensemble | 460 | 563 | 760 | 311 | 334 | 771 | 897 | 1092 |
| | | (509 to 629) | (672 to 841) | | | | (843 to 963) | (1006 to 1175) |
| NASA-GISS | 467 | 511 | 841 | 42 | 40 | 509 | 551 | 881 |
| GFDL-ESM2M ensemble | 452 | 839 | 1240 | 309 | 425 | 761 | 1264 | 1665 |
| | | (747 to 919) | (1166 to 1373) | | | | (1172 to 1344) | (1475 to 1798) |
| GFDL-ESM4 | 467 | 925 | 1227 | / | / | / | / | / |
| IPSL ensemble | 465 | 841 | 1049 | 72 | 92 | 537 | 913 | 1141 |
| | (465 to 465) | (839 to 843) | (1033 to 1065) | | | (537 to 537) | (931 to 935) | (1125 to 1157) |
| MIROC | 461 | 641 | 966 | / | / | / | / | / |
| MPI | 467 | 845 | 1146 | 204 | 293 | 671 | 1138 | 1439 |
| NorESM2 | 461 | 491 | 1088 | / | / | / | / | / |
| Bern ensemble | 459 | 604 | 899 | 110 | 128 | 669 | 732 | 1027 |
| | | (356 to 867) | (615 to 1259) | | | | (484 to 995) | (743 to 1387) |
| UVic | 472 | 765 | 1021 | 103 | 173 | 575 | 938 | 1194 |
| Multi-model mean | 462 | 685 | 1002 | 170 | 215 | 632 | 901 | 1185 |
| Multi-model range | (452 to 472) | (491 to 925) | (760 to 1240) | (41 to 310) | (39 to 425) | (509 to 770) | (550 to 1264) | (881 to 1665) |

**Table 5.** Cumulative $G_{ATM}$, $S_{OCEAN}$ and $S_{LAND}$ sinks in 2020 and 2150. $S_{LAND}$ values can only be diagnosed for models which have provided an estimate of their simulated $E_{LUC}$ (see Methods).

| Models | $G_{ATM}$ (PgC) | | | $S_{OCEAN}$ (PgC) | | | $S_{LAND}$ (PgC) | | |
|---|---|---|---|---|---|---|---|---|---|
| | 2020 | 2150 1.5°C | 2150 2.0°C | 2020 | 2150 1.5°C | 2150 2.0°C | 2020 | 2150 1.5°C | 2150 2.0°C |
| CESM2 | 308 | 172 | 284 | 166 | 257 | 330 | 207 | 383 | 492 |
| EC-Earth ensemble | 354 (348 to 360) | 231 (205 to 262) | 318 (269 to 363) | 209 (207 to 210) | 328 (304 to 347) | 391 (362 to 411) | 220 (214 to 227) | 345 (332 to 354) | 392 (376 to 400) |
| NASA-GISS | 272 | 204 | 369 | 160 | 292 | 400 | 80 | 57 | 113 |
| GFDL-ESM2M ensemble | 309 (299 to 313) | 375 (340 to 401) | 582 (532 to 661) | 197 (196 to 198) | 451 (426 to 478) | 557 (540 to 582) | 254 (250 to 261) | 435 (400 ; 467) | 521 (504 to 553) |
| GFDL-ESM4 | 331 | 392 | 550 | 206 | 426 | 509 | / | / | / |
| IPSL ensemble | 256 (256 to 256) | 295 (294 to 296) | 370 (362 to 378) | 123 (123 to 123) | 318 (314 to 322) | 378 (374 to 382) | 128 (128 to 128) | 312 (311 to 313) | 404 (399 to 409) |
| MIROC | 236 | 212 | 349 | 171 | 316 | 406 | / | / | / |
| MPI | 299 | 330 | 475 | 166 | 348 | 433 | 214 | 468 | 540 |
| NorESM2 | 292 | 140 | 416 | 183 | 271 | 424 | / | / | / |
| Bern ensemble | 290 (282 to 301) | 241 (137 to 347) | 375 (245 to 557) | 160 (152 to 166) | 300 (227 to 387) | 398 (299 to 507) | 122 (118 to 126) | 191 (121 to 263) | 254 (175 to 340) |
| UVic | 293 | 349 | 488 | 161 | 371 | 455 | 124 | 213 | 245 |
| Multi-model mean | 294 | 267 | 416 | 173 | 335 | 426 | 168 | 300 | 370 |
| Mulit-model range | (236 to 354) | (139 to 354) | (284 to 354) | (128 to 208) | (256 to 450) | (330 to 556) | (79 to 254) | (56 to 468) | (113 to 540) |

## 6.2 Regional distribution

The continuous ocean carbon uptake until the end of the simulations is limited to specific regions (Fig. 8a,c,d). While the ocean carbon sink increases almost everywhere from the early 20[th] century to the mid-21[st] century, it only continues to take up carbon after temperature stabilization at the end of the 21[st] century in the Southern Ocean and the low latitude regions close to the equator. Especially the Southern Ocean around 60ºS remains a prominent and enduring carbon sink post temperature stabilization, a consistent feature across models (no stippling in Fig. 8a,d). Cumulatively, the Southern Ocean south of 30ºS,

representing 35 % of the ocean area, takes up 42 % (35 % to 46 %) of the global ocean carbon uptake by 2020, rising to 46 % (35 % to 57 %) by 2150 in the 2.0 ºC simulation shown here. This region remains a sink until 2300 for the models that have run long simulations (GFDL-ESM2M, MIROC, Bern, not shown). Another strong present-day carbon sink (panel c), the subpolar North Atlantic, ceases to absorb carbon when the surface temperature stabilizes (panel d). The subpolar North Atlantic north of 40ºN (and using the northern boundary of Fay and McKinley (2014)) represents 3 % of the ocean area, but takes up 7 % (3

% to 11 %) of global ocean carbon uptake by 2020, a fraction that decreases to 5 % (1 % to 11 %) by 2150. The prevalence of the Southern Ocean carbon sink is consistent with results from CMIP5 and CMIP6 simulations for the historical period and for idealized 1pctCO$_2$ experiments (Frölicher et al., 2015; Terhaar et al., 2021; Williams et al., 2023). The pronounced long-term steady carbon sink in the Southern Ocean can be attributed to the high carbon concentration feedback and efficient surface to deep export of anthropogenic carbon shown in earlier studies (Tjiputra et al., 2010; Roy et al., 2011). In the mid-latitude

subtropics, some models even simulate less carbon uptake or outgassing due to the accumulation of anthropogenic carbon near the surface layers (Couespel and Tjiputra, 2024; Rodgers et al., 2020).

The land carbon uptake also persists only in particular regions during temperature stabilization (Fig. 8b,e,f). Some strong and robust carbon sink regions in the present era (2011-2030), particularly in tropical areas like South America, central Africa and part of Indonesia, experience a complete cessation of carbon uptake after 2100. However, strong terrestrial carbon uptake

persists in the Northern Hemisphere during temperature stabilization, including regions like Central America, temperate and boreal forests in Eastern North America and Eurasia.

Compared to the ocean carbon uptake, the land carbon uptake has locally a larger uncertainty across models, although much of this uncertainty is located in regions where no carbon exchange is found during the pre-industrial period (e.g. desertic regions), and is thus an artefact of the method and therefore not indicated by stipples on the maps (see caption in Fig. 8).

The land and ocean carbon sink patterns look qualitatively similar as previous results based on low emission scenarios (Canadell et al., 2021). The novelty of these simulations is a quantification of these sinks for a given temperature level and the representation of carbon cycle dynamics beyond the 21[st] century under strong mitigation scenarios. For example, here we show that the North Atlantic ocean sink vanishes during and after stabilization while the Southern Ocean remains an active sink. The divergence in carbon uptake between these regions can be attributed to distinct ocean circulation patterns. The Southern

Ocean is the region where old, anthropogenic CO$_2$-poor circumpolar deep waters are being upwelled (e.g. Mikaloff Fletcher et al., 2006), even during global surface climate stabilization. This upwelling results in continuous carbon uptake due to the positive air-sea gradient in anomalous $p$CO$_2$ (Frölicher et al., 2015). On the other hand, when global temperature stabilizes, the

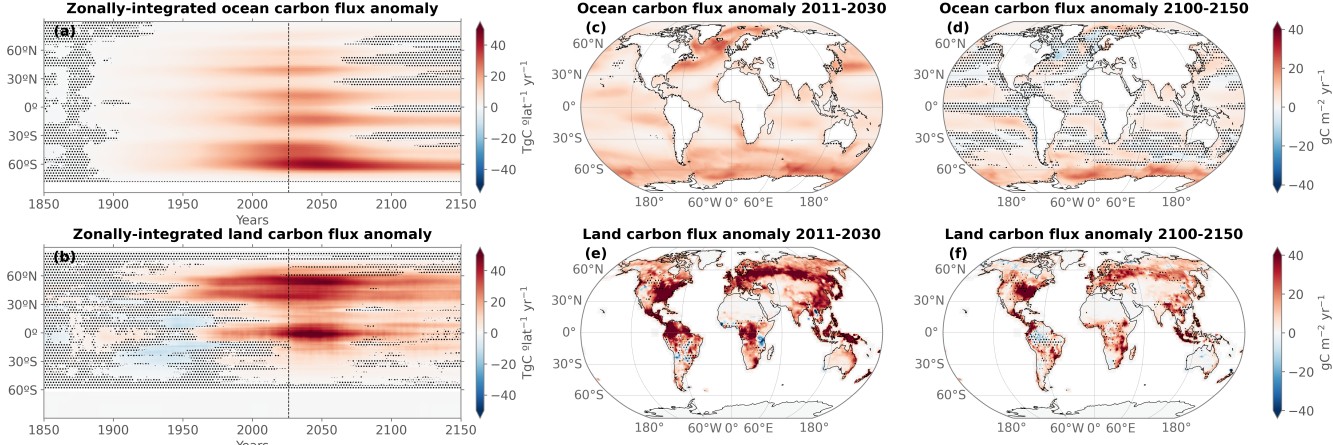

**Figure 8.** Multi-model mean ocean (top, $S_{OCEAN}$) and land (bottom, $S_{LAND} - E_{LUC}$) carbon flux anomalies in the 2.0 °C simulation relative to 1850-1900. In panels (a) and (b) the fluxes are zonally-integrated and smoothed with a 31-year running mean, while panels (c)-(f) display maps averaged between 2011-2030 and 2100-2150. Stipples indicate where less than 80 % of the models agree on the sign of the anomaly. Stippling is not shown where carbon fluxes during the 1850-1900 period are lower than 2.5 gC m$^{-2}$ yr$^{-1}$.

Atlantic Meridional Overturning Circulation (AMOC) is expected, at least in some models, to recover after an initial decrease (e.g. Manabe and Stouffer, 1994; Sigmond et al., 2020; Frölicher et al., 2020; Schwinger et al., 2022; Bonan et al., 2022), a phenomenon associated with a recovery of deep convection in the subpolar North Atlantic, enhanced mixing of anthropogenic $CO_2$-rich waters previously sequestered at depth, and a consequently lower ocean carbon uptake in this region. A stronger AMOC would also transport more warm surface waters with high anthropogenic carbon concentrations to the high latitudes where the anthropogenic carbon would be outgassed during cooling (Siegenthaler and Joos, 1992; Völker et al., 2002; Tjiputra et al., 2010), a phenomenon already simulated for the present time in the Arctic Ocean while atmospheric $CO_2$ is still increasing (Terhaar et al., 2020). The exact drivers of the vanishing Atlantic Ocean carbon sink under temperature stabilization will be the subject of future dedicated studies using the AERA-MIP simulations.

## 7 Discussion

The AERA-driven simulations achieve temperature stabilization with realistic $CO_2$-fe and $CO_2$ emission pathways across an ensemble of comprehensive Earth System Models. However, applying the AERA with ESMs is more complex than traditional concentration-driven simulations and comes with its caveats and limitations.

We robustly quantify a decrease in $CO_2$-fe emissions towards very low emissions for the selected 1.5 °C and 2.0 °C warming levels. To which extent reductions in $CO_2$-fe emissions are realized by reductions in emissions of fossil fuel $CO_2$ vs. other agents depends on scenario choices. We opted for the low emission high mitigation scenario SSP1-2.6 for all greenhouse gases except $CO_2$, and for anthropogenic aerosols. While the scenario choice does not impact initial estimates of the REB and $CO_2$-fe

emission pathways (Terhaar et al., 2022a), the choice does affect fossil fuel $CO_2$ emissions, atmospheric $CO_2$ and the land and ocean carbon sinks (Terhaar et al., 2023). Thus, the quantitative estimates presented here for the carbon budget in Section 4, carbon distribution in the Earth System and atmospheric $CO_2$ offer a likely pathway under low emissions and high mitigation but may not encompass all uncertainties. These estimates may change if alternative trajectories for non-$CO_2$ radiative agents and land use change are pursued in the future (Terhaar et al., 2023). Nevertheless, qualitative statements remain robust, such as the necessity to substantially reduce fossil fuel $CO_2$ emissions for temperature stabilization and the continuous carbon uptake by the Southern Ocean even under temperature stabilization.

The choice of the scenario for non-$CO_2$ radiative agents can also lead to unwanted trajectories of the $CO_2$ emission curve if the prescribed non-$CO_2$ scenario is not ideal for the chosen temperature level. In the case of the 2 °C warming level, fossil fuel $CO_2$ emissions increase from 2025 to 2030 and start to decrease strongly afterwards (Fig. 4b). The temporal increase in fossil fuel $CO_2$ emissions is a result of a strong decline in $CO_2$-fe emissions from declining non-$CO_2$ radiative agents under SSP1-2.6, which exceeds the necessary total $CO_2$-fe emission decline calculated by the AERA in 2025. As a result, fossil fuel $CO_2$ emissions increase to compensate for the strong decline in $CO_2$-fe emissions from declining non-$CO_2$ radiative agents. Such unwanted effects could be avoided by adjusting the SSP1-2.6 scenario so that mitigation efforts of non-$CO_2$ agents and land use start later, as already applied by Millar et al. (2017), or simply by choosing the best fitting non-$CO_2$ SSP scenario for each temperature level. Another possibility which might be offered in the future when ESMs develop an emission-driven mode for non-$CO_2$ agents is to scale the $CO_2$ and non-$CO_2$ emissions, as already applied in Terhaar et al. (2022a) with an EMIC.

The non-$CO_2$ and land use change emissions do not only add uncertainties to the carbon budget but also add uncertainties to the AERA at each stocktake. At each stocktake, the AERA needs information about the $CO_2$-fe emissions from non-$CO_2$ radiative agents and from land use change. Unfortunately, most models do not provide emissions from land use change or the non-$CO_2$ radiative forcing from which the corresponding $CO_2$-fe emissions can be estimated. As a result, the $CO_2$-fe emissions from land use change and from non-$CO_2$ agents provided to AERA are estimated from other sources (see Methods), which lead to discrepancies between the real $CO_2$-fe emissions in the model and the estimated $CO_2$-fe emissions seen by the AERA. This discrepancy affects the estimate of the TCRE-fe and the future $CO_2$ emission curve, which is estimated as the difference between the AERA-derived $CO_2$-fe emission curve and the $CO_2$-fe emissions from land use change and non-$CO_2$ radiative agents. Fortunately, the simulations provided here with an ensemble of ESMs demonstrate that the re-evaluation of the TCRE-fe and the future $CO_2$-fe emission curve every 5 years allows all models but one to reach the prescribed temperature level without any larger divergence (Fig. 2). For future simulations, possibly within CMIP7, diagnosing each models' effective radiative forcing of non-$CO_2$ agents and land use change (Smith, 2020; Zelinka et al., 2023) would likely lead to even more precise results, as described in more detail for the IPSL model in Appendix A. This is particularly important for the aerosol radiative forcing, which is the most uncertain component, with important repercussions for the ability to reach climate goals (Watson-Parris and Smith, 2022).

Not only the implementation of the AERA with ESMs introduces uncertainties, but the comprehensive emission-driven, fully coupled Earth system models themselves also have limitations. For example, the models do not include changes in freshwater input from melting ice sheets and the associated impacts on ocean circulation. This additional freshwater input

can modulate global and regional temperature and carbon cycle responses (e.g., Bronselaer et al. (2018); Li et al. (2023)). However, ice sheet changes are expected to occur on timescales over many centuries, potentially beyond the time horizon of largest reduction in $CO_2$-fe emissions. Nevertheless, the AERA approach would adjust allowable emissions in response to emerging, unforeseen feedbacks when applied to real-world emission and temperature data (Terhaar et al., 2022a). Additionally, most models in our study either neglect or poorly represent permafrost dynamics and often underestimate soil carbon stocks

in the northern high latitudes (MacDougall et al., 2015; Burke et al., 2017; Gasser et al., 2018; Lowe and Bernie, 2018; Burke et al., 2020; MacDougall, 2021). Permafrost thaw due to global warming has the potential to release a substantial amount of carbon stored in soil for millennia into the atmosphere over a relatively short period (e.g., Schuur et al. (2015)). Therefore, $CO_2$-fe emission pathways that account for the release of permafrost soil carbon may differ from those shown here, potentially requiring more stringent emissions reduction to achieve the prescribed global warming levels. Future model studies

incorporating permafrost soil carbon are necessary to quantify this effect and fully capture the uncertainty in land and ocean carbon-warming feedbacks. Another caveat is the large uncertainties in the non-$CO_2$ GHG warming feedbacks (Stocker et al., 2013; Battaglia and Joos, 2018; IPCC, 2021), which are currently not represented in our approach with prescribed non-$CO_2$ forcing. These feedbacks would potentially reduce the emission budget over time and potentially make negative emissions over long time periods necessary to maintain a stable temperature (Palazzo Corner et al., 2023). Another limitation of Earth

System Model lies in the long equilibration timescales, resulting in potential drift in carbon fluxes, amongst other variables. A correction is often applied by removing the pre-industrial trend, but this is not taken into account in the current version of the AERA, as it would add another layer of complexity to the simulation setup.

     Using CMIP6 models, Arora et al. (2020) demonstrated that models that included a representation of the terrestrial nitrogen cycle typically show lower carbon uptake than those without, because of widespread nitrogen limitation of the vegetative carbon

sink. This finding has been corroborated by the ZECMIP analysis (MacDougall et al., 2020). A lower terrestrial carbon sink in models with a terrestrial nitrogen cycle implies stronger reductions in total $CO_2$-fe emissions in those models. This is indeed simulated in both the 1.5 °C and 2.0 °C global warming level scenarios (see Figure B8). For example, under the 1.5 °C global warming scenario, the multi-model mean $CO_2$-fe emissions in 2050 for the six models including a terrestrial nitrogen cycle are -1.6 PgC yr$^{-1}$, compared to 2.0 PgC yr$^{-1}$ for the 5 models without it. Averaged between 2100-2150, the $CO_2$-fe emissions

are 0.8 PgC yr$^{-1}$ for models with a terrestrial nitrogen cycle and 1.5 PgC yr$^{-1}$ without. The minimum $CO_2$-fe emissions in the 1.5 °C scenario between 2026 and 2150 are -1.6 PgC yr$^{-1}$ in the models with a nitrogen cycle, and 1.3 PgC yr$^{-1}$ in the models without. The emission budget between 2020 and 2150 is also affected when partitioning the models into these two groups. The $CO_2$ only budget is 550 (250 to 950) Gt$CO_2$ for the models with a nitrogen cycle in the 1.5 °C scenario (median and 17[th]-83[rd] percentile range), and goes up to 1500 (1000 to 1850) Gt$CO_2$ for the models without. In the 2.0 °C scenario, the budget is

1850 (1250 to 2500) Gt$CO_2$ for the models with a nitrogen cycle and 2300 (2000 to 3100) Gt$CO_2$ for the models without. The analysis indicates that part of the spread in $CO_2$-fe emission pathways and budgets can potentially be attributed to whether models account for the nitrogen limitation in the terrestrial carbon sink. However, the model spread in $CO_2$-fe emissions also emerges due to other model differences, such as in ocean carbon uptake (e.g., Terhaar et al. (2022b)) or climate sensitivity (Cox et al., 2018).

## 8 Conclusions

This study presents multi-ESM emission-driven projections compatible with internationally-agreed climate goals. We showed that the Adaptive Emission Reduction Approach (AERA) proposed by Terhaar et al. (2022a) works not only for EMICs but also for more complex, higher-resolution fully-coupled ESMs. The prescribed temperature levels are reached even when the simulated non-$CO_2$ radiative forcing and land use change emissions are not fully known, with the exception of one ESM. The success of the AERA-MIP simulations across a large group of modeling centres shows that the AERA can be used in subsequent model intercomparison projects, such as CMIP7, with many applications for these temperature stabilization simulations.

Unlike the standard CMIP scenarios that simulate different warming levels for the same prescribed $CO_2$ concentration or emission pathways (Tebaldi et al., 2021), all models converge to a given warming level. The convergence to a common warming level with varying emissions now allows to quantify the diversity of model responses in terms of emission budgets and pathways compatible with these warming levels, as well as resulting atmospheric $CO_2$ levels, carbon cycle responses, and their effect on ecosystems, such as ocean acidification (Terhaar et al., 2023). While globally integrated results are qualitatively similar to previous results with EMICs (Terhaar et al., 2022a, 2023; Goodwin et al., 2018b), the ESM simulations here allow to explore a more quantitative and regional focus, to better quantify uncertainties, and especially to quantify the importance of internal climate variability.

To limit warming to 1.5 ℃ or 2.0 ℃ (i.e., an allowable warming of 0.28 ℃ and 0.78 ℃ from year 2020 forward), drastic reductions in greenhouse gas emissions are necessary. If such immediate and drastic emission reductions of around -1 to -2 PgC yr$^{-2}$ were implemented, both $CO_2$-fe and fossil fuel $CO_2$ emissions may even be allowed to stay positive (on the order of 1 PgC yr$^{-1}$ upon stabilization). However, the amount of allowed continuous $CO_2$-fe emissions after the temperature is stabilized is strongly model-dependent, with a large spread found across ESMs. The large spread is mainly caused by varying zero emissions commitments and responses to non-$CO_2$ forcing agents across the model ensemble.

Unlike the remaining carbon budget concept, which assumes constant TCRE and a constant ZEC centered around zero, the AERA simulations provide the realized emission budget until the temperature reaches the prescribed warming level and stabilizes. This modelling exercise is a powerful tool to explore the evolution of the TCRE and the associated budgets as the climate responds to emissions mitigation. Indeed, the spread in the zero emissions commitment in combination with a non-constant TCRE-fe results in an initially biased estimate of the remaining emission budget. On average across models, the REB that was estimated in 2025 could change by a factor of 1.4 to 2 (Table 3). The direction of change mainly depends on the ZEC, with a negative ZEC allowing for more emissions. Here, the REB was on average underestimated in line with a slightly negative multi-model mean ZEC (MacDougall et al., 2020). However, there are large uncertainties around the ZEC and hence the development of carbon and emission budgets with time.

A few ensemble simulations pointed to a significant role of internal variability, potentially explaining 30 % to over 50 % of the inter-model spread in compatible emissions, atmospheric $CO_2$ levels and emission budgets. The origin of this uncertainty partly lies in the estimate of the anthropogenic warming, which in practise differs among members and can lead to large differences in the emission budgets, as also pointed out by Tokarska et al. (2020). This indicates that caution is required when

interpreting remaining budgets estimates, and planning for a margin of error in mitigation pathways to avoid overshooting the desired warming level.

In addition to the "relative warming level" simulations presented here (i.e., same amount of remaining warming for all models after 2020 based on observations), the AERA can also be used to make simulations with an "absolute warming level" (i.e. all models are set up to warm by the same amount relative to 1850-1900) or with an overshoot. Absolute warming level simulations allow to explore climate impacts of global surface temperature stabilization at different warming levels across Earth System Models (King et al., 2021), which will be the focus of future dedicated studies. Temperature overshoot simulations with the AERA allow to define the magnitude and length of the overshoot by varying prescribed temperature levels over time, e.g., a first warming level of 2.0 °C until 2050 followed by a step-wise reduction of that level every 5 years to 1.5 °C in 2100 (Terhaar et al., 2022a).

The AERA framework proposed here accounts for prescribed global surface temperature levels, aligned with international climate agreements. However, other climate change impacts, such as ocean acidification, sea level rise, interior ocean changes, terrestrial productivity, regional extremes, pose important risks for ecosystem and human societies. Extending the AERA to other targets, and towards avoiding crossing some of the Earth's planetary boundaries (Rockström et al., 2009), as proposed by Steinacher et al. (2013); Seneviratne et al. (2016); Avrutin et al. (2023), would enable to constrain the emission budget and pathways towards a safer world.

*Code and data availability.* The AERA code is distributed as a python module openly available under https://github.com/Jete90/AERA, with a guided documentation and examples. The AERA-MIP model outputs used in this study are available under Silvy et al. (2024). The Python code used to produce the figures of this paper will be made openly available upon final publication.

## Appendix A: Further details on the AERA configuration of the participating models

All participating models are listed in Table A1. Both HadCM3-FaIR2 and MPI performed the AERA simulations with a previous version of the AERA code using an impulse response function to the radiative forcing to estimate anthropogenic temperature instead of the 31-year running mean (as in Terhaar et al. (2022a)). The HadCM3-FaIR2 configuration is described in details in Lee et al. (in review). Briefly, it uses the Hadley Centre Coupled Model version 3 (HadCM3, Collins et al. (2001)), with 29 members from a physics perturbed parameter ensemble (Sparrow et al., 2018). Because HadCM3 runs in concentration-driven mode and does not solve the carbon cycle, it is coupled to the Finite amplitude Impulse Response (FaIR) version 2 (Leach et al., 2021) at each AERA stocktake. The FaIR parameters are chosen to fit each member of the HadCM3 ensemble based on the 1881-2025 simulation period. The carbon cycle component of FaIR is used both to derive the $CO_2$-fe emissions from non-$CO_2$ agents, and to convert the $CO_2$ emissions given by AERA to $CO_2$ concentration to prescribe to HadCM3 every 5 years.

**Table A1.** Earth System Models of full and intermediate complexities participating in AERA-MIP.

| Model full name | Abbreviation | References | Simulated years | Ensemble members |
|---|---|---|---|---|
| ACCESS-ESM1-5 | ACCESS | Ziehn et al. (2020) | 1850-2200 | 1 |
| CESM2 | CESM2 | Danabasoglu et al. (2020) | 1850-2200 | 1 |
| EC-Earth3-CC | EC-Earth | Döscher et al. (2022) | 1850-2169 | 3 |
| GFDL-ESM2M | GFDL-ESM2M | Dunne et al. (2012, 2013) | 1861-2300 | 5 |
| GFDL-ESM4 | GFDL-ESM4 | Dunne et al. (2020) | 1850-2150 | 1 |
| IPSL-CM6-LR-ESMCO2 | IPSL | Boucher et al. (2020) | 1850-2150 | 2 |
| MIROC-ES2L | MIROC | Hajima et al. (2020) | 1850-2300 | 1 |
| MPI-ESM1-2-LR | MPI | Mauritsen et al. (2019) | 1850-2150 | 1 |
| NASA-GISS-E2-1-G-CC | NASA-GISS | Kelley et al. (2020); Ito et al. (2020); Miller et al. (2021); Lerner et al. (2024) | 1850-2150 | 1 |
| NorESM2-LM | NorESM2 | Seland et al. (2020); Tjiputra et al. (2020) | 1850-2200 | 1 |
| Bern3D-LPX | Bern | Ritz et al. (2011) | 1850-2300 | 72 |
| UVic-ESCM-2.10 | UVic | Mengis et al. (2020) | 1850-2200 | 1 |
| HadCM3-FaIR2 | HadCM3-FaIR2 | Lee et al. (submitted) | 1881-2100 | 29 |

The time series prescribed to AERA (default or model-estimated $E_{LUC}$ and $E_{non\text{-}CO_2\text{-}fe}$) are listed in Table A2. To estimate their own simulated $E_{LUC}$, most models compare land-air carbon fluxes between two concentration-driven simulations following historical+SSP1-2.6 $CO_2$ concentrations (1850-2100), one with land use change activated and another without (Lawrence et al., 2016; Liddicoat et al., 2021). This difference is then smoothed with a 21-year running mean to remove large interannual variations. Eight models prescribed their internally-estimated $E_{LUC}$ to AERA. The NASA-GISS model underestimates $E_{LUC}$ as it estimated the emissions from land use change due to crop cover change only, but did not include the transport of crop harvest from land, and did not include the deforestation component of $E_{LUC}$. UViC used an estimate based on the carbon flux from vegetation burning only. The time series of all $E_{LUC}$ estimates prescribed to AERA are shown in Fig. A1a (same as Fig. 4c).

In the analysis of carbon distribution (Equation 6) presented in the paper, we corrected and re-estimated some of these $E_{LUC}$ time series, to obtain the best estimate possible of the internally-simulated $E_{LUC}$. These time series are shown in Fig. A1b. GFDL-ESM2M performed a posteriori the concentration-driven simulations with and without land use change activated, providing its estimate of $E_{LUC}$ for the carbon analysis from 1861 to 2100. We then extended the time series to 2300 by applying a linear decay to zero emissions from 2100 to 2150, and maintaining the emissions at zero afterwards. We similarly corrected the MPI estimate between 2100 and 2150 by linearly decaying the emissions to zero. Dedicated UVic simulations were additionally performed to better estimate $E_{LUC}$ in the AERA simulations for both prescribed warming levels. IPSL was also able to diagnose online the simulated $E_{LUC}$ emissions within the AERA simulations. For CESM2, we were able to estimate $E_{LUC}$ for the historical period, using the available concentration-driven *hist* and *hist-nolu* simulations available on the

**Table A2.** AERA forcings prescribed to each model.

| Model | $E_{LUC}$ provided to AERA | $E_{non\text{-}CO_2\text{-}fe}$ provided to AERA |
|---|---|---|
| ACCESS-ESM | Model estimate | Default |
| CESM2 | Default | Default |
| EC-Earth | Model estimate | Default |
| GFDL-ESM2M | Default | Model estimate |
| GFDL-ESM4 | Default | Model estimate from GFDL-ESM2M |
| IPSL-ESM | Default/Model estimate | Default/Model estimate |
| MIROC-ES2L | Default | Default |
| MPI-ESM | Model estimate | Default |
| NASA-GISS | Model estimate | Model estimate |
| NorESM2 | Default | Default |
| Bern3D-LPX | Model estimate | Default |
| UVic-ESCM | Model estimate | Default |
| HadCM3-FaIR2 | Model estimate | Model estimate |

Earth System Grid Federation and the method described above (Lawrence et al., 2016). However, the extension for the SSP1-2.6 scenario was not available. Nonetheless, the $E_{LUC}$ term during the historical period matched the default $E_{LUC}$ forcing from the Bern model adjusted time series (not shown), so we used the default estimate for the carbon distribution analysis.

The simulated effective radiative forcing from non-$CO_2$ agents was diagnosed in dedicated simulations for IPSL, but only for $CH_4$, $N_2O$ and aerosols, which have major climate impacts amongst non-$CO_2$ forcing agents. For other agents, we used the estimates from Smith et al. (2023) for the historical period and from Smith (2020) for the SSP1-2.6 scenario period post-2022.

For the IPSL model, one additional set of simulations were performed by prescribing the internally-calculated $E_{non\text{-}CO_2\text{-}fe}$ and $E_{LUC}$ emissions, enabling us to test the effect of the mismatch in $E_{non\text{-}CO_2\text{-}fe}$ and $E_{LUC}$ between the default time series

and the internally-simulated emissions. The member with the internally-calculated emissions allowed for a shorter temperature undershoot in the 1.5 ℃ simulation, and a better temperature stabilisation within the 2.0 ℃ uncertainty range in the associated simulation. However, minor differences in emission budget and cumulative fossil fuel emissions were found overall between these two ensemble members, due to compensating effects between $CH_4$ and $N_2O$ in the estimated and the internally-calculated radiative forcing, and between $E_{non\text{-}CO_2\text{-}fe}$ and $E_{LUC}$. As an indication, the $EB_{2026}(2025)$ (i.e., the REB from beginning of

year 2026 estimated at the first stocktake) differs by 3 PgC for the 1.5 ℃ warming level and by 15 PgC for the 2.0 ℃ level between the two ensemble members. The differences increase to 52 PgC, respectively 48 PgC, for $EB_{2026}(2100-2150)$.

The simulations performed with the Bern3D-LPX model correspond to the configuration in Terhaar et al. (2023), where two parameters (ocean heat uptake efficacy and feedback parameter) were varied to obtain 9 values of Earth Climate Sensitivity (ECS) spanning the range 2.23 ℃ to 4.63 ℃. Since Bern3D-LPX does not represent atmospheric variability on GSAT, a

synthetic noise was added on the GSAT output read by AERA for each of these 9 ECS values, providing 8 synthetic ensemble

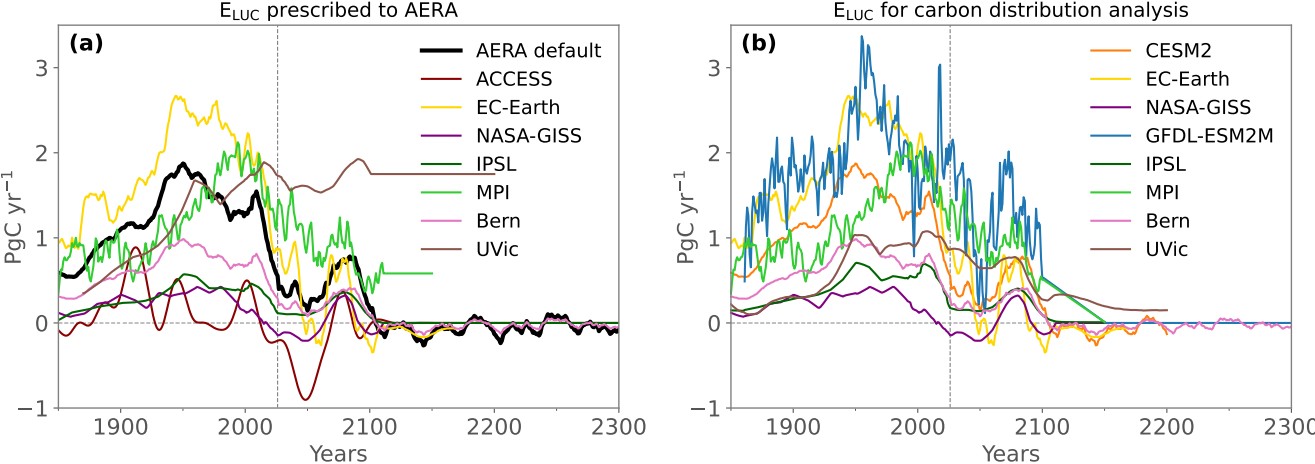

**Figure A1.** $E_{LUC}$ diagnosed in the models and default AERA input. Panel a) shows the time series prescribed to AERA and panel b) the corrected time series used in the carbon distribution analysis of this paper (see text for details of the corrections).

members per configuration, hence a total of 72 ensemble members. This Bern3D-LPX range across members is thus more representative of parametric uncertainty, and not comparable to the perturbed initial-condition ensembles of EC-Earth and GFDL-ESM2M.

## Appendix B: Additional figures

*Author contributions.* TLF, JT and FJ designed the AERA-MIP protocol. YS, FAB, FL and JT participated in the testing and validation of the protocol. YS gathered and analyzed the simulations. YS, JT and TLF wrote the first draft of the manuscript. JT performed the Bern3D-LPX simulations. FAB and FL performed the GFDL-ESM2M simulations. JRB performed the CESM2 simulations. RF, VL and ET performed the EC-Earth3-CC simulations. PC performed the IPSL-CM6-LR-ESMCO2 simulations. MD and TL performed the ACCESS-ESM1-5 simulations. GG performed the MPI-ESM1-2-LR simulations. TH and MK performed the MIROC-ES2L simulations. NYK, PL and AR

performed the NASA-GISS-E2-1-G-CC simulations. DL performed the HadCM3-FaIR2 simulations. NM and EAM performed the UVic-ESCM-2.10 simulations. JS and JTJ performed the NorESM2-LM simulations. ES performed the GFDL-ESM4 simulations. DP provided the GFDL-ESM2M radiative forcing from non-CO₂ radiative agents. All authors contributed to discussions and final writing of the manuscript. TLF led the revision of the paper and YS performed the additional analyses.

*Competing interests.* All authors declare no competing interests.

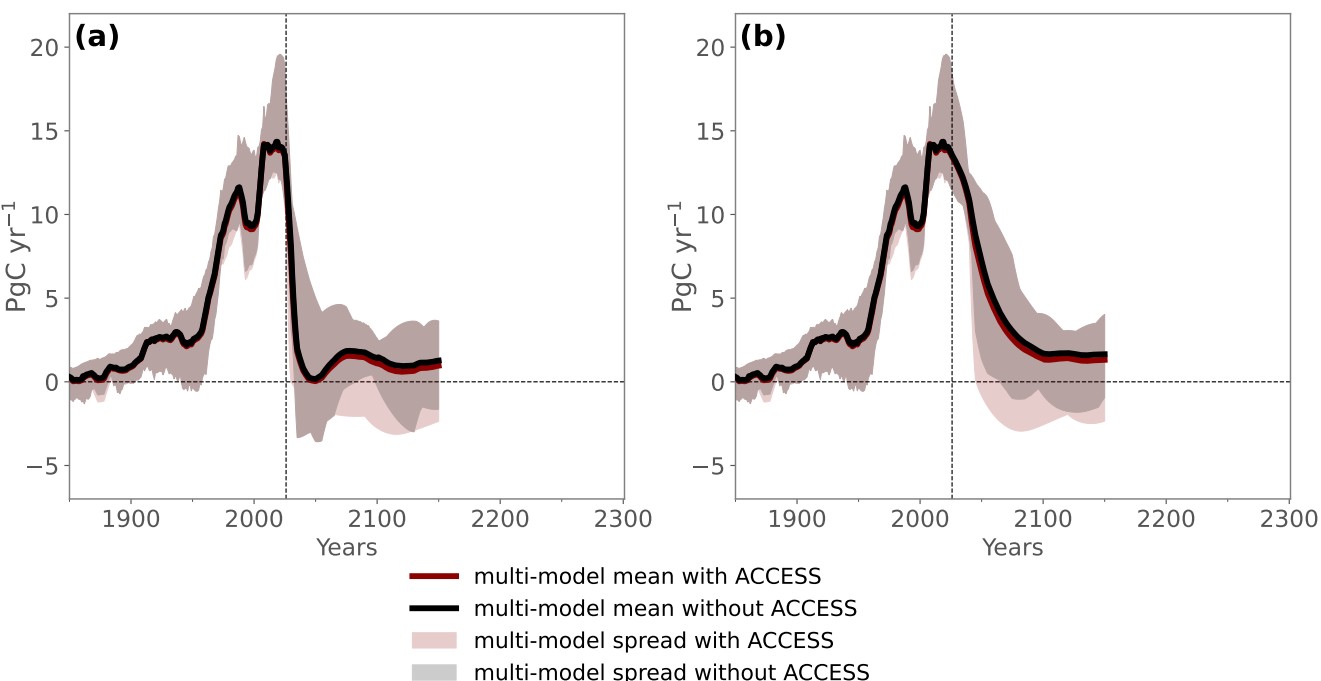

**Figure B1.** Simulated multi-model mean $CO_2$-fe emissions for the 1.5°C and 2.0°C warming levels with and without including the ACCESS model.

*Acknowledgements.* This work was supported by the European Union's Horizon 2020 research and innovation program under grant agreement no. 821003 (project 4C). The work reflects only the authors' view; the European Commission and their executive agency are not responsible for any use that may be made of the information for the work. TLF (project Overshoot) and FJ (project Peakcarbon) also thank the CSCS Swiss National Supercomputing Centre for computing resources. TLF (PP00P2_198897), JT (PZ00P2_209044), and FJ (200020_200511) have received funding from the Swiss National Science Foundation. JT has also received funding from the Woods Hole

Oceanographic Institution Postdoctoral Scholar Program. EAM and NM are funded under the Emmy Noether scheme by the German Research Foundation 'FOOTPRINTS – From carbOn remOval To achieving the PaRIs agreemeNt's goal: Temperature Stabilisation' (ME 5746/1-1). GPP received funding from the European Union's Horizon Europe Research and Innovation Programme under grant agreement No 101081179 (DIAMOND). TZ receives funding from the Australian Government under the National Environmental Science Program (NESP). TH and MK were supported by the MEXT-Program for advanced studies of climate change projection (SENTAN, Grant Number

JPMXD0722681344). TH was also supported by the Environment Research and Technology Development Fund (JPMEERF21S20820) of the Environmental Restoration and Conservation Agency provided by the Ministry of Environment of Japan. AR, PL, and NYK were supported by the NASA Modeling, Analysis and Prediction program, with computing resources provided by the NASA High-End Computing (HEC)

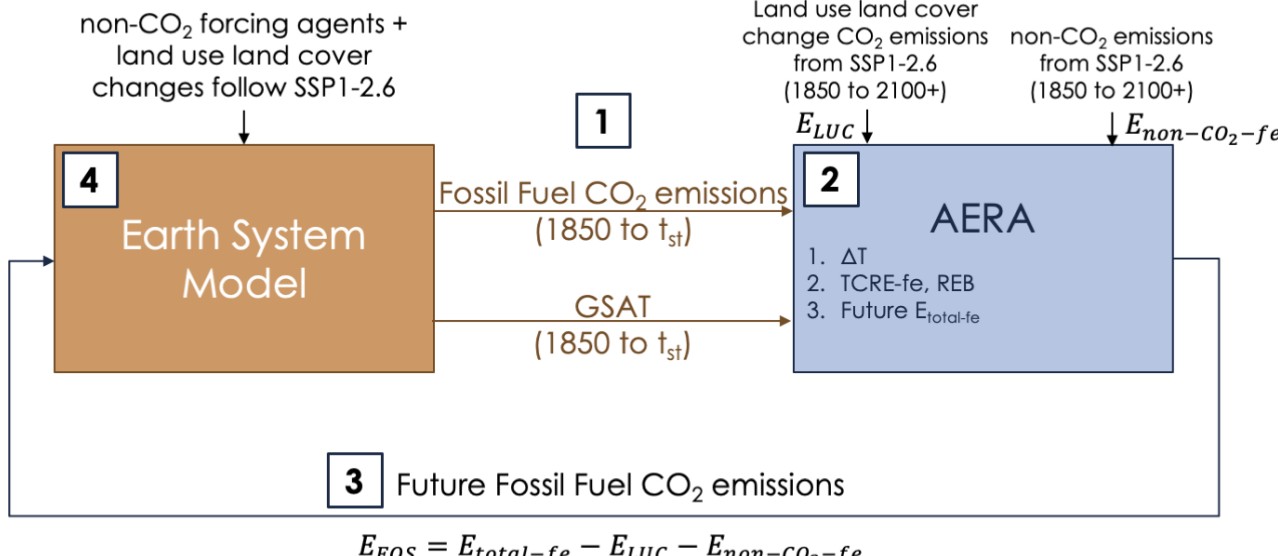

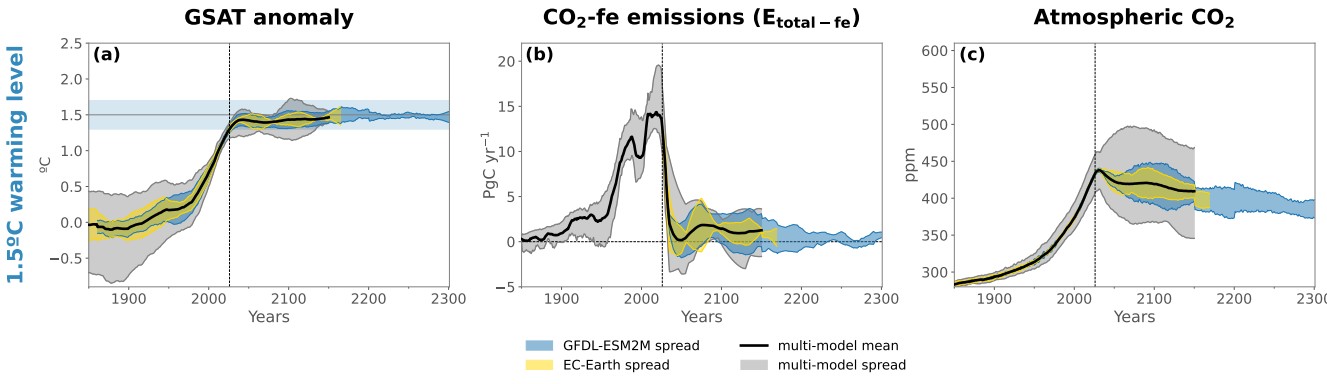

**Figure B2.** Schematic of the feedback loop between the Earth System Model and the AERA module. Steps 1 to 4 are performed at each stocktake year, starting at the end of 2025.

**Figure B3.** Same as Fig. 3 but for the 1.5 ℃ warming level.

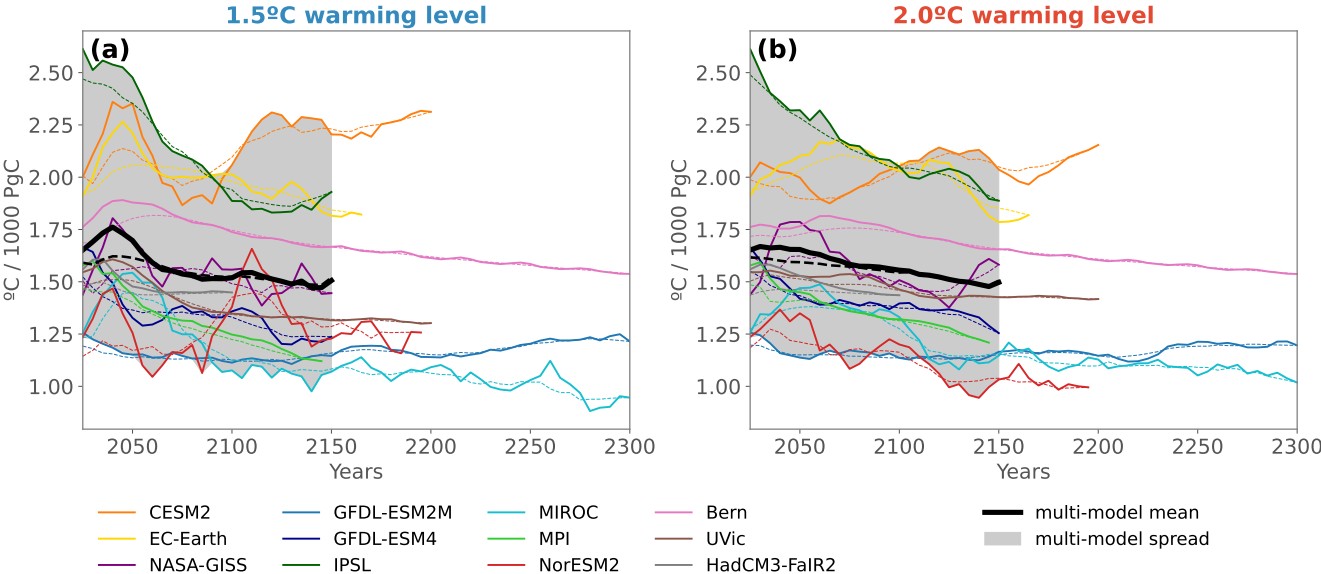

**Figure B4.** Solid lines show TCRE-fe calculated by AERA at each stockake year. Dotted lines indicate the TCRE-fe calculated a posteriori with the final ("true") estimate of the 31-year running mean time series of GSAT. The deviation between the solid and dotted lines occurs because the temperature fit becomes less accurate when the temperature increase slows toward stabilization. As the warming rate decreases, the 31-year mean will also decrease. Therefore the "true" estimate of the 31-year mean will be smaller than the estimate at each stocktake. Consequently, the TCRE will be smaller than estimated at the stocktake during periods of slowing down temperature increases. In addition, the reconstructed TCRE-fe (dotted lines) is less variable than the TCRE-fe calculated by AERA, which calculates $\Delta T$ based on the last year of the extended 31-year running mean time series until each stocktake year, which introduces some noise compared to the true final estimate.

Program through the NASA Center for Climate Simulation (NCCS) at Goddard Space Flight Center. VB was supported by the European Research Council (ERC) under the European Union's Horizon 2020 research and innovation programme (grant agreement number 951288).

**2026 CO$_2$-fe Emission Budget (EB$_{2026}$)**

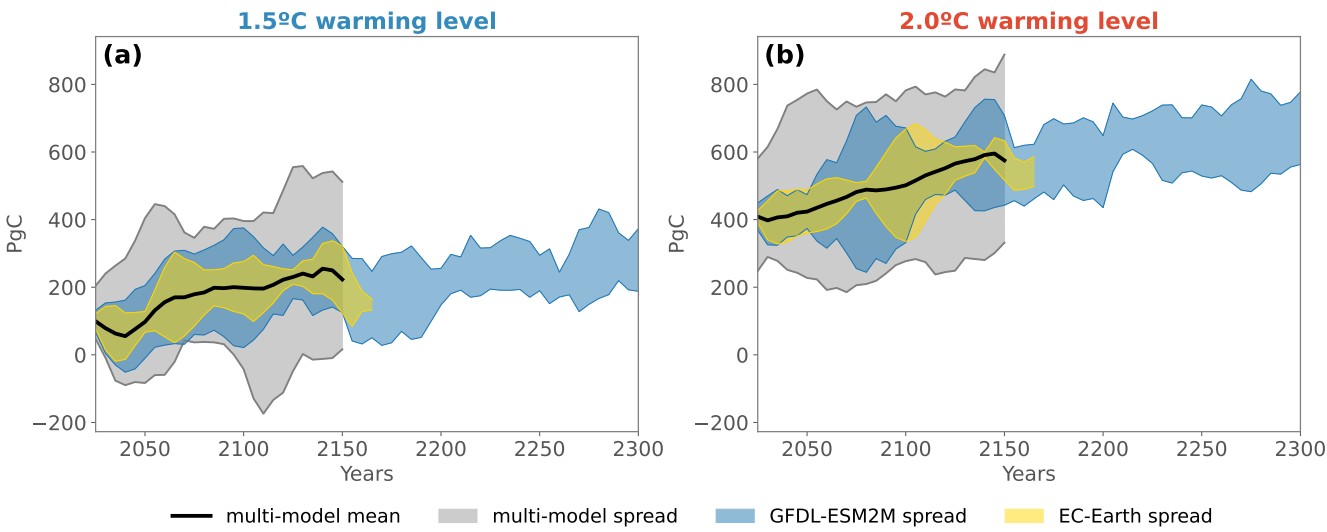

**Figure B5.** Same as Fig. 5 but showing only the multi-model mean and spread, as well as the min-max range across members for the GFDL-ESM2M and EC-Earth ensembles, centered on the multi-model mean.

$EB_{2026}([2100-2150]) - EB_{2026}(2025)$

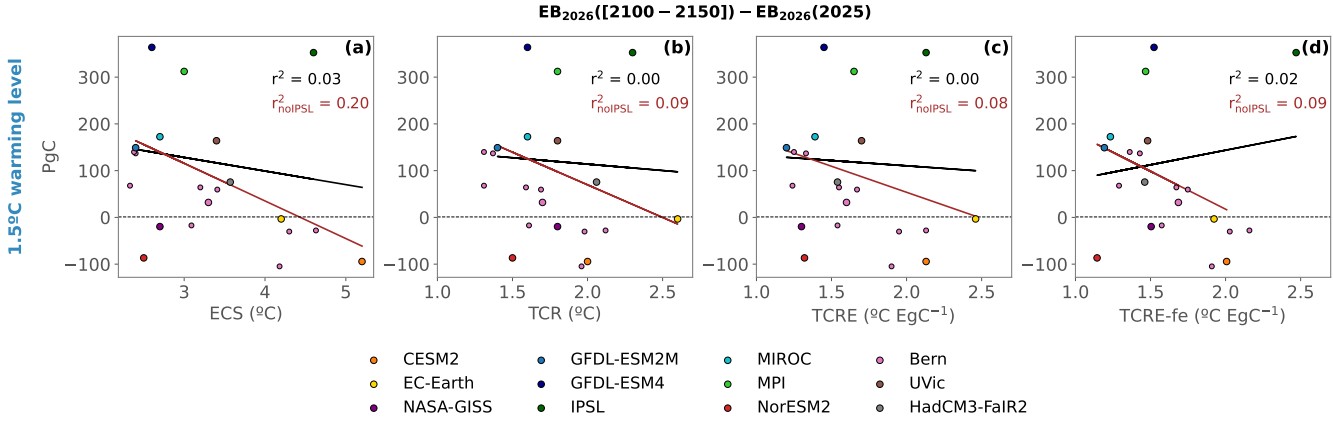

**Figure B6.** Same as Fig. 6 but for the 1.5 ºC warming level.

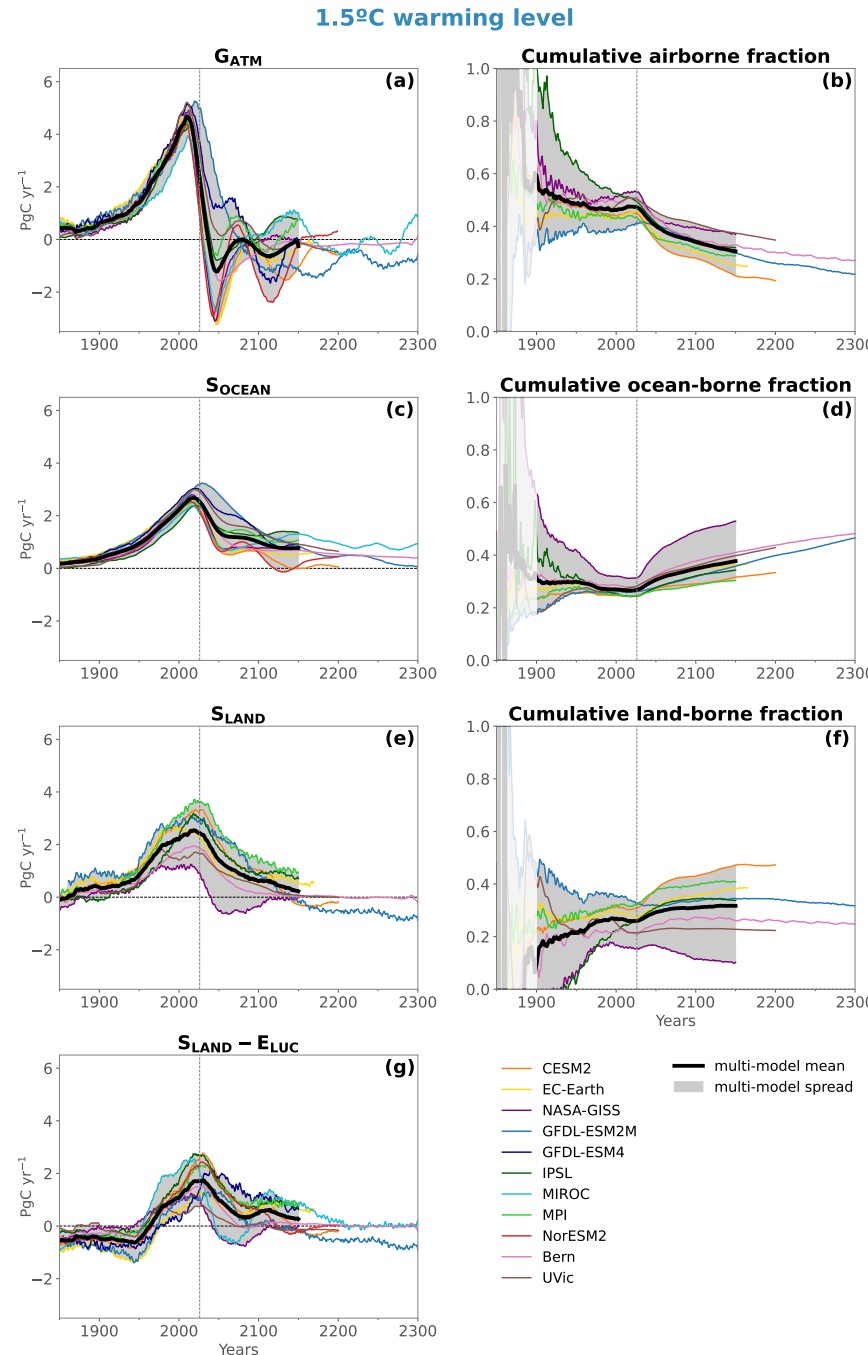

**Figure B7.** Same as Fig. 7 but for the 1.5 ºC warming level.

## 1.5ºC warming level    2.0ºC warming level

## CO$_2$-fe emissions (E$_{total - fe}$)

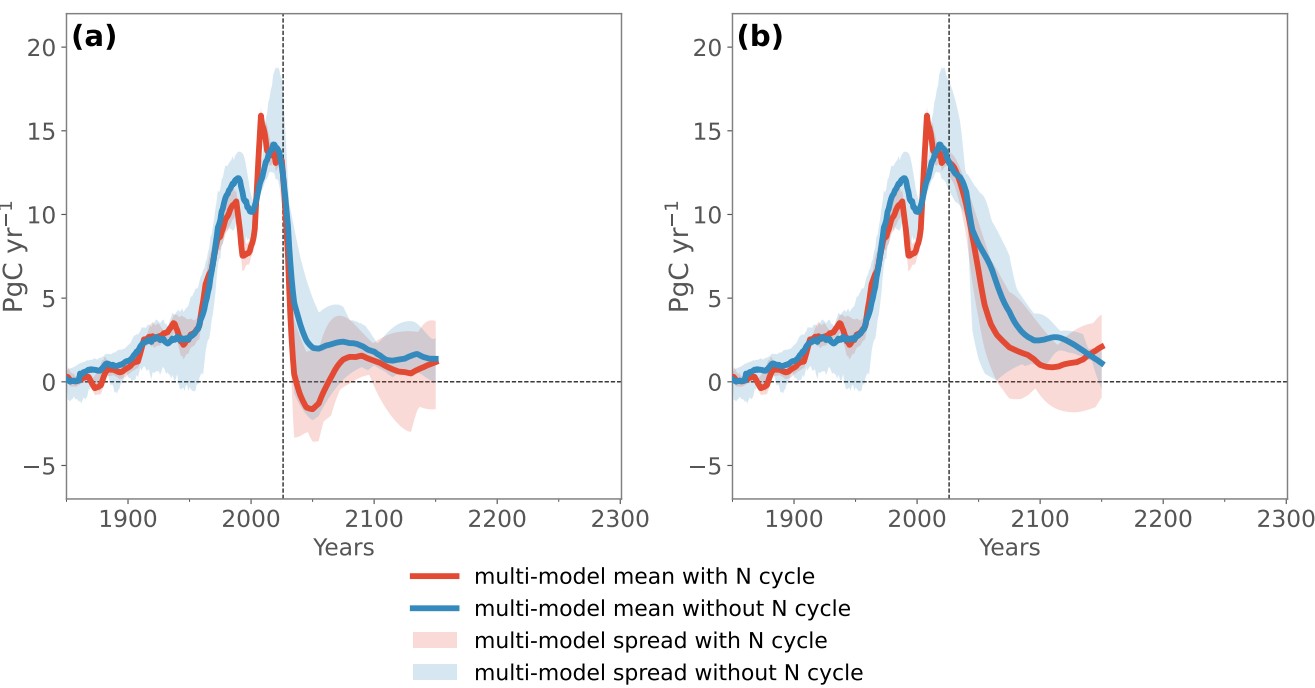

**Figure B8.** Simulated multi-model mean CO$_2$-fe emissions for the 1.5 °C and 2.0 °C warming levels for the models that include a terrestrial nitrogen cycle (red; CESM2, EC-Earth, MIROC, MPI, NorESM2, Bern) and for the models that do not include a terrestrial nitrogen cycle (blue; NASA-GISS, GFDL-ESM2M, GFDL-ESM4, IPSL, UVic). ACCESS is not included.

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
