# Peer review of "AERA-MIP: Emission pathways, remaining budgets and carbon cycle dynamics compatible with 1.5 °C and 2 °C global warming stabilization"

_EGUsphere, 2024_

## Author Comment (AC1)

**Response to the editor and reviewers**
We thank the editor and the three reviewers for the critical assessment of our work and their very helpful and constructive comments. We have addressed all comments point by point and revised our manuscript accordingly.

**Reviewer 1**
Review of: AERA-MIP: Emission pathways, remaining budgets and carbon cycle dynamics compatible with 1.5 ºC and 2 ºC global warming stabilization

Overall Evaluation:

The paper presents the results of AERA-MIP, an effort to use Earth system models to quantify the emission pathway needed to reach global temperature stabilization. I commend the authors for the substantial effort needed to put together a model intercomparison of this complexity, and for this comprehensive manuscript. I have some relativity minor suggestions to clarify some of the methods and results.

We thank the reviewer for the positive assessment of our manuscript.

General Comments:

(1) I recommend that you add either a box or an appendix to explain CO2 forcing equivalent emissions to a more general Climate Science audience. Model intercomparison papers such of these are often read by audiences far beyond those immediately involved in the field, and without explaining CO2 forcing equivalent emissions that audience is going to be very confused. It is also important to explain if there is an assumed airborne fraction built into CO2 forcing equivalent emissions.

We appreciate the reviewer's suggestion to include a box explaining the $CO_2$-fe emissions concept. However, we haven't found any examples of papers in ESD that contain such a box. Therefore, we defer to the editor's decision on whether this is permissible. If it is not possible, we will incorporate the text as an additional paragraph in section 2.1, rather than placing it in the appendix, as it might be overlooked there. The new paragraph (or box) reads as follows:

"*The concept of $CO_2$-fe emissions is used to unify the emissions of various radiative forcing agents and precursors into a single metric (Jenkins et al. 2018, Allen et al. 2018, Smith et al. 2021). $CO_2$-fe emissions for all non-$CO_2$ agents represent the $CO_2$ emissions that would produce the same radiative forcing trajectory as these non-$CO_2$ emissions. While cumulative $CO_2$ emissions largely determine anthropogenic warming, non-$CO_2$ radiative forcing agents such as methane, nitrous oxide and aerosols also play an important role. Quantifying the impact of these non-$CO_2$ agents on global temperatures is complicated by existing methodologies, which often use conventional Global Warming Potentials or other metrics to convert the non-$CO_2$ radiative forcing agents into 'CO_2-equivalent' emissions. The $CO_2$-fe emissions framework offers an alternative, well-suited for comparing emissions from different agents in the context of temperature stabilization pathways (Terhaar et al. 2022). It also offers an opportunity to compare emission reductions of different radiative forcing agents for ecosystem impacts (Terhaar et al. 2023). The $CO_2$-fe emissions from non-$CO_2$ agents, $E_{non-CO2-fe}$, are estimated based on the radiative forcing time series of non-$CO_2$ agents (Smith et al. 2021):*

$$E_{non-CO2-fe}(t) = \frac{1}{\alpha} \left(\frac{dF_{non-CO2}(t)}{dt} + \rho F_{non-CO2}(t)\right)$$

*where $F_{non-CO2}$ is the radiative forcing of non-CO₂ agents, $\rho$ is the rate of decline in radiative forcing under zero emission over decadal to centennial timescales (0.33%), and $\alpha$ is a constant representing the forcing impact of ongoing $CO_2$ emissions (1.08 Wm⁻² per 1000 GtC). By varying the relative shares of $CH_4$ and $N_2O$ emissions and radiative forcing from aerosols in the total $CO_2$-fe emissions, Terhaar et al. (2022) demonstrated the robustness of the $CO_2$-fe approach in translating contributions from different radiative forcing agents into $CO_2$-fe emissions.*"

(2) There are three explanations in the paper as to why many of the models project that continued fossil fuel CO2 emissions could be compatible with temperature stabilization:

I) Negative ZEC.

II) Net-negative forcing from non-CO2 forcing being compensated with fossil fuel CO2 emissions.

III) Net-negative carbon emissions from land use change being compensated with fossil fuel CO2 emissions.

The paper shows that all three mechanisms are at work at different times for at least some models. It would be good to collect these explanations together in the discussion to clarify what are the dominant mechanisms on what time-scale and how mechanism may translate to the natural world.

To emphasize the contributions of the fossil-fuel and non-CO₂ radiative forcing agents as well as land use changes, we have included a new figure in the main text that shows the CO₂-fe emissions from all potential sources that are either simulated by the model or prescribed to AERA. The new figure illustrates that CO₂ emissions from land use changes and CO₂-fe emissions from non-CO₂ agents are likely not the primary reasons for the continued positive total CO₂-fe emissions after temperature stabilization.

We have extended section 3.3 (Emissions from fossil fuel, non-CO₂ agents and land use change) as follows:

"*The prescribed CO₂-fe emissions from non-CO₂ agents (Figure 3d; colored lines for models that were able to estimate their internal radiative forcing; black line for the others; see Methods) exhibit a rapid decrease after year 2026, reaching maximum negative emissions around year 2054 of -1.4 PgC yr⁻¹ and stabilizing at slightly positive levels of 0.2 PgC yr⁻¹ after 2100 for the default prescribed emissions (black line). CO2 emissions from land use change (Figure 3c) generally remain slightly positive throughout the 21ˢᵗ century, except in the NASA-GISS, EC-Earth and ACCESS models, where they are temporarily negative. Post-2100, the land-use change CO₂ emissions prescribed to AERA stabilize around zero in all models. Both the non-CO₂ and the land-use change CO₂-fe emissions follow identical trajectories for the 1.5°C and 2.0°C global warming level simulations, as they follow the SSP126 scenario for both warming levels.*

*It is important to mention that the simulated model-specific CO2 emissions from land use change and CO2-fe emissions from non-CO2 agents are not available as output from all models. Nevertheless, negative CO2-fe emissions from non-CO2 radiative forcing agents are only possible if the radiative forcing from these agents is decreasing (see equation above). However, the radiative forcing agents follow the SSP126 or RCP26 scenario and after 2100 they are set to constant values for all models suggesting that substantial negative non-CO2 radiative forcing is unlikely post-2100. In addition, land-use is also set to be constant after 2100. Considering that $CO_2$ emissions from both land use change and non-$CO_2$ agents are likely to be zero or slightly above zero after temperature stabilization, it is reasonable to conclude that the positive CO2-fe and fossil fuel CO2 emissions after stabilizing warming are more likely a consequence of the overall negative ZEC rather than net negative forcing from non-CO2 forcing or from land use change."*

[Figure]

*Figure 3: Simulated fossil fuel $CO_2$ emissions, emissions from land use change and non-$CO_2$ forcing agents, and atmospheric $CO_2$. The multi-model mean, excluding the ACCESS model, is displayed by the black thick line, with the grey shading indicating the min-max spread. The*

*ensemble mean is shown for models that have several ensemble members. The vertical dotted line at the year 2026 marks the beginning of the AERA simulations. In panels c and d, the black line shows the default AERA input for both warming levels, whereas the colored lines show the diagnosed land-use change or non-CO₂ forcing equivalent emissions for models that do not use AERA default values.*

As a result of the revisions, figure 2 of the main text has also been slightly modified. It now only shows the temperature and CO₂-fe emissions response.

[Figure]

*Figure 2: Simulated temperature anomaly CO2-fe emissions and fossil fuel CO2 emissions for the 1.5°C and 2.0°C warming levels. Panels a) and b) display the 31-year running mean of the global surface air temperature (GSAT) anomaly, aligned with the observed value in 2020. Panels c) and d) illustrate CO2-fe emissions. The multi-model mean, excluding the ACCESS model, is displayed by the black thick line, with the grey shading indicating the min-max spread. The ensemble mean is shown for models that have several ensemble members. The vertical dotted line at year 2026 marks the beginning of the AERA simulations.*

(3) Related to comment 2, I suggest adding a section on anthropogenic carbon emissions similar to section 3.3. That is, I think it is also important to highlight what the sum of fossil fuel and land use change emissions to distinguish between true continued net CO2 emissions and fossil fuel emissions being compensated by net-negative land use change emissions.

The combined CO2 emissions from fossil fuels and land use change are only marginally different from fossil fuel alone, as land use change CO2 emissions after 2026, especially during the stabilization phase, are minimal. Please see Figure R1 for the combined CO2 emission from fossil fuel and land use change. As we have already included a new Figure 3

with land use change CO2 emissions in the main text, along with a paragraph describing land-use change emissions (refer to the reply to the previous comment), we do not include Figure R1 in the manuscript.

[Figure]

*Figure R1: Simulated CO2 emissions from fossil fuel and land use change for the 1.5°C and 2.0°C warming levels.*

(4) When examining the model's carbon cycle responses you should examine whether models with a terrestrial nitrogen cycle have a different response.

We have now added a paragraph to the discussion section and also included a new Figure in the Appendix.

*"Using CMIP6 models, Arora et al. (2020) demonstrated that those models with a terrestrial nitrogen cycle typically show lower carbon uptake than those without, because of widespread nitrogen limitation of the vegetative carbon sink. This finding has been corroborated by the ZECMIP analysis (MacDougall et al. (2020)). A lower terrestrial carbon sink in models with a terrestrial nitrogen cycle implies stronger reductions in $CO_2$-fe emissions in those models. This is indeed simulated in both the 1.5°C and 2.0°C global warming level scenarios (see Figure A4). For example, under the 1.5°C global warming scenario, the multi-model mean CO2-fe emissions in 2050 for the six models including a terrestrial nitrogen cycle are -1.6 PgC yr$^{-1}$, compared to 2.0 PgC yr$^{-1}$ without it. Averaged between 2100-2150, the CO2-fe emissions are 0.8 PgC yr$^{-1}$ for models with a terrestrial nitrogen cycle and 1.5 PgC yr$^{-1}$ without. The minimum CO2-fe emissions in the 1.5°C scenario between 2026 and 2150 are -1.6 PgC yr$^{-1}$ in the models with a nitrogen cycle, and 1.3 PgC yr$^{-1}$ in the models without. The analysis indicates that part of the spread in CO2-fe emissions pathways can potentially be attributed to whether models include a terrestrial nitrogen cycle. However, the model spread in CO2-fe emissions also emerges due to other model differences, such as in ocean carbon uptake (e.g., Terhaar et al. 2022) or climate sensitivity (Cox et al. 2018)."*

[Figure]

Figure A4: Simulated multi-model mean $CO_2$-fe emissions for the 1.5°C and 2.0°C warming levels for the models that include a terrestrial nitrogen cycle (red; CESM2, EC-Earth, MIROC, MPI, NorESM2, Bern) and for the models that do not include a terrestrial nitrogen cycle (blue; NASA-GISS, GFDL-ESM2M, GFDL-ESM4, IPSL, UVic). ACCESS is not included.

Specific Comments:
Line 19: I suspect that only a handful of your models had a representation of the permafrost carbon feedback (UVic, CESM2, maybe Bern?), so this statement needs a caveat.

We have excluded the northern high-latitude land sink from the abstract. The sentence now reads: "*On the other hand, the Southern Ocean remains a carbon sink over centuries after temperatures stabilize.*" We also included a new caveat paragraph in the discussion section that addresses permafrost carbon: "*Additionally, most models in our study either neglect or poorly represent permafrost dynamics (Burke et al. 2020) and often underestimate soil carbon stocks in the northern high latitudes. Permafrost thaw due to global warming has the potential to release a substantial amount of carbon stored in soil for millennia into the atmosphere over a relatively short period (e.g., Schuur et al. 2015). Therefore, CO2-fe emissions pathways that account for the release of permafrost soil carbon may differ from those shown here, potentially requiring more stringent emissions reduction to achieve the prescribed global warming levels. Future model studies incorporating permafrost soil carbon are necessary to quantify this effect and fully capture the uncertainty in land and ocean carbon-warming feedbacks.*"

Line 41: Delete 'Damon'.

Deleted.

Figure 1: Delete 'or Concentration'. This panel of the figure is clearly only represents an emissions pathway. The 'or Concentration' is just confusing.

Agree and deleted.

Line 53: 'climate-carbon feedbacks' are either part of TCRE if they are included in the ESM or are part of the 'unrepresented feedbacks' if not represented in the ESM. Delete 'climate-carbon feedbacks'.

Deleted.

Section 2.2. Should note here that the two models with the largest positive ZEC values in ZECMIP, UKESM and CNRM, did not participate in AERA-MIP.

We did not include this sentence because 4 out of the 10 ESMs participating in AERA-MIP did not participate in ZECMIP (i.e., EC-Earth3-CC, GFDL-ESM4, IPSL-CM6-LR-ESMCO2, NASA-GISS-E2-1-G-C). These models may have similarly large positive ZEC values. Adding such a statement thus appears speculative.

Line 128 to 129: Fix grammar.

Changed to: "*Ideally, more ensemble members are necessary to properly quantify the internal variability (Lehner et al. 2020).*"

Line 132 to 136: Is this due to the way non-CO2 is simulated in the radiative transfer code of ACCESS, or due to something like dynamic CH4 emissions?

We do not know the exact cause of this mismatch. This would need to be explored in future studies.

In response to Referee 3, we have now included a figure in the Appendix (Figure A3) that incorporates the ACCESS model in the multi-model mean. The new figure demonstrates that our study's results are qualitatively robust and do not depend on whether the ACCESS model is included in the ensemble statistics or not.

Line 180: Fix grammar.

Changed to: "*Further details on the configuration of the AERA simulations are provided in Appendix A*".

Line 205: I suggest not using abbreviations as variables. For consistency with your other variables $C_{AF}$, $C_{OF}$, and $C_{LF}$ would be preferable.

Changed as suggested.

Line 210: Operationally these are actually sums not integrals (just adding up the yearly or monthly values from the model output). Would be sensible just to use sum notation to not make this seem more complicated than it really is.

Following reviewer #2 suggestion, we added 'dt' to the integrals instead of using sums as suggested here.

Line 218 to 220: Fix grammar.

Changed to: "*The IPSL model temporarily leaves the 1.5°C uncertainty range, and MIROC briefly leaves the 2.0°C uncertainty range. In the 1.5°C simulation, the multi-model mean GSAT anomaly enters the target uncertainty range in 2026, i.e., the year when the first AERA period begins (black thick line in Fig. 2a)*."

Line 317 to 324: Be clear which numbers are calendar years in this paragraph.

We now write: "*..of 53 ppm in year 2025..*", "*..After year 2025..*", " *..by year 2050..*", "*..by year 2100..*", and "*..by year 2150..*".

Line 334 to 335: 'already' should go before 'exhibits'.

We removed the word 'already'.

Line 379 to 380: Be clear that you are referring to either effective-TCRE or TCRE-fe. By definition true-TCRE only includes the effect of CO2 emissions.

We clarified that we mean TCRE-fe.

Line 392: Cannot make a strong statement of significance for your collection of ESMs. Climate models are not independent, and mathematically closer to phylogenetically related (Knutti et al. 2013). Independence is a foundational assumption underlying all tests of significance. Can replace 'significant' with 'strong' or 'substantial'.

Changed to 'strong' as suggested.

Figure 6: Would be good to include a panel with the land-use change emissions used. Maybe as its own figure as land-use change is a forcing not an output of the model experiment.

The land-use change emissions are now shown in Figure 3 (previously in Appendix Figure A2a). Although, each model simulates its own land-use change emissions based on the prescribed land-use change, these emissions are not available for all models so that land use change emissions are prescribed in the AERA based on estimates from the global carbon budget (see Methods or Terhaar et al. (2022)).

Line 443 to 445: Re-write this sentence for clarity.

Changed to "*From 1850 to 2020, 294 PgC (ranging from 236 to 354 PgC) of the total cumulative CO2 emissions from fossil fuels (462 PgC) and land use change (170 PgC) remained in the atmosphere. During the same period, the ocean has taken up 173 PgC (ranging from 128 to 208 PgC), and the land absorbed 168 PgC (ranging from 79 to 254 PgC)(Table 4)*."

References:

Knutti R, Masson D, Gettelman A. Climate model genealogy: Generation CMIP5 and how we got there. Geophysical Research Letters. 2013 Mar 28;40(6):1194-9.

Sincerely:

-Andrew MacDougall

---

## Author Comment (AC2)

**Response to the editor and reviewers**
We thank the editor and the three reviewers for the critical assessment of our work and their very helpful and constructive comments. We have addressed all comments point by point and revised our manuscript accordingly.

**Reviewer 2**
The study estimates emissions pathways that are adjusted to be compatible with temperature targets, referred to as Adaptive Emission Reduction Approach (AERA) using AERA-MIP, made up of 13 full Earth system models, 2 intermediate complexity Earth system models and 1 ocean general circulation model coupled to a carbon cycle emulator. The analyses of AERA-MIP are reported here and provide compatible emission pathways and remaining carbon budgets, together with ocean and land carbon responses. The responses are shown for model means and the model ranges, as well as including ensemble means and ensemble ranges to reveal internal variability where appropriate. The study is very comprehensive, thorough and impressive.

We thank the reviewer for the positive assessment of our manuscript.

I only have minor comments, but given the importance of the study I recommend that these comments are taken on board to aid communication:

1. There is a statement (L177) that the future CO2 emissions curve compatible with a temperature target is largely insensitive to the non-CO2 radiative forcing and land-use changes. As long as this statement is representative of the study (as this seems slightly surprising to me), then I recommend making more of this statement and discussing the implications in the Conclusion and perhaps including in the Abstract.

There seems to be a misunderstanding, as we refer to $CO_2$-fe emissions in the text, not $CO_2$ emissions. The $CO_2$-fe emissions are largely insensitive to the chosen land use and non-$CO_2$ forcings, as demonstrated by Terhaar et al. (2022), Jenkins et al. (2018), Allen et al. (2018), and Smith et al. (2021). We have extended our description of CO2-fe emissions in the Methods (or in a new box) to avoid misunderstanding. We do not wish to reiterate this finding in the abstract of this paper.

2. The methodology is clearly described and novel, but does overlap with a prior study of Goodwin et al. (2018a) introducing "Adjusting Mitigation Pathways". The present approach does extend the prior study of Terhaar et al. (2022a). The text states the limitations of the Goodwin et al. (2018a) approach, but does not outline the similarity in the two approaches. Further explanation would be helpful to the wider community.

We modified the text to:

"*Such adaptive approaches have been proposed and tested with reduced complexity models running forward from the present day, and offer promising potential. Goodwin et al. (2018a) introduced the 'Adjusting Mitigation Pathways" method using a climate box model. In this approach, the remaining carbon budget to a predefined warming target is estimated by using the near-linear relationship between warming and cumulative carbon emission from a historically forced simulation. Then the remaining budget is distributed in the future and reassessed every ten years.*"

and..

*"Even though the philosophy of AERA is similar to the Adjusting Mitigation pathways approach (Goodwin et al. 2018a), AERA provides smoother emission pathways, incorporates non-$CO_2$ agents, always stabilizes at the warming targets within +/- 0.2°C, and can also be applied to run simulations that temporarily overshoot the warming target."*

3. Please include any caveats about the set of Earth system models that are included in the study, so that there is clarity about any limitations to the approach. For example, there is probably a large range in the climate feedbacks in your model set and limitations in the closures of some carbon cycles particularly for the land, which might then affect the estimates of the remaining carbon budget.

We have added the following caveat paragraph to the discussion section:

*"Even though our analysis is based on comprehensive emission-driven, fully coupled Earth system models, a few caveats need to be discussed. First, the models do not include changes in freshwater input from melting ice sheets and the associated impacts on ocean circulation. This additional freshwater input can modulate global and regional temperature and carbon cycle responses (e.g., Bronselaar et al. 2018, Li et al. 2023). However, ice sheet changes are expected to occur on timescales over many centuries, potentially beyond the time horizon of largest reduction in CO2-fe emissions. Nevertheless, the AERA approach would adjust allowable emissions in response to emerging, unforeseen feedbacks when applied to real-world emission and temperature data (Terhaar et al. 2022). Additionally, most models in our study either neglect or poorly represent permafrost dynamics (Burke et al. 2020) and often underestimate soil carbon stocks in the northern high latitudes. Permafrost thaw due to global warming has the potential to release a substantial amount of carbon stored in soil for millennia into the atmosphere over a relatively short period (e.g., Schuur et al. 2015). Therefore, CO2-fe emissions pathways that account for the release of permafrost soil carbon may differ from those shown here, potentially requiring more stringent emissions reduction to achieve the prescribed global warming levels. Future model studies incorporating permafrost soil carbon are necessary to quantify this effect and fully capture the uncertainty in land and ocean carbon-warming feedbacks. Another caveat is the large uncertainties in the non-$CO_2$ GHG warming feedbacks (Canadell et al. 2021), which are currently not represented in our approach with prescribed non-$CO_2$ forcing."*

4. While the manuscript is very comprehensive, the text is cryptic in places and sometimes not easy to follow due to the large number of acronyms and the choices made of those acronyms. I recommend making variable names more internally consistent, rather than using a range of different symbols to represent variables with the same units. For example, the carbon inventory in (4) includes variables written as E, G and S with subscripts, but when those variables are referred to later in isolation their meaning requires the reader to go back and find their definitions (such as L319). Likewise Table 2 uses REM and EB_2026 and both are in PgC, so unsure as to why the change in symbols used. Sometimes repetition in their definitions would also be helpful to the reader, such as in the final Discussion or Conclusions. I also recommend including a Table to list those variables.

We enhanced the readability of the text by carefully reviewing the manuscript, making slight revisions and reducing the use of acronyms. When an acronym has not been used recently, we now introduce its full name. For example, the sentence on lines 318-320 now reads: *"… as all models have identical prescribed fossil fuel $CO_2$ emissions.."*. We have retained the usage of REB and EB, as REB is commonly employed in literature to refer to the remaining emissions budget of today. However, in the manuscript, we recalculated today's emissions budget based on simulations until 2100-2150. To clearly differentiate between these two emissions budget we opted for two distinct acronyms.

Following our revisions, we no longer believe it necessary to include a table listing the individual variables.

In summary, this study is very impressive and substantial, and will make an important contribution to discussions about the remaining carbon budget, and provide key information for the next IPCC report. Minor editorial work can help in the readability of the study for a wider audience

Minor details

Line 41 Add Goodwin et al. (2018b)

Added.

Equation (6) and (7) add the dt in the integrals to make more explicit.

Added.

Line 243 Overlong sentence including repeated use of but.

Changed to: "*The positive emissions and large model spread during the temperature stabilization phase are consistent with the overall negative but highly uncertain multi-decade temperature response after zero CO2 emissions across a range of EMICs and ESMs (MacDougall et al. 2020; Jenkins et al. 2022b).*"

Line 506. Try to avoid using "it", you know what you mean, but better to be explicit to the reader.

The sentence now reads: "*While the scenario choice does not impact initial estimates of the REB and CO2-fe emission pathways (Terhaar et al. 2022a), the choice does affect fossil fuel CO2 emissions, atmospheric CO2 and the land and ocean carbon sinks (Terhaar et al. 2023).*"

---

## Author Comment (AC3)

**Response to the editor and reviewers**

We thank the editor and the three reviewers for the critical assessment of our work and their very helpful and constructive comments. We have addressed all comments point by point and revised our manuscript accordingly.

**Reviewer 3**

Comments on "AERA-MIP: Emission pathways, remaining budgets and carbon cycle dynamics compatible with 1.5oC and 2oC global warming stabilization" by Silvy et al.

Overall, I think this is a really nice paper and an important contribution that shows a new way of using ESMs to understand remaining emissions budgets at different reference warming levels. The authors have done a great job of describing a complex set of experimental results. I do have a couple concerns that I describe below, but overall I think it is a very worthwhile publication.

We thank the reviewer for the positive assessment of our manuscript.

My first concern is semantic but I think it is important. I object to the usage of the word "target" in the sense of "global warming target" starting in the very first sentence of manuscript, and its many instances thereafter. The Paris agreement (English-language version) uses the word "target" only in article 4 to describe emissions reduction targets, it never refers to warming levels as "targets". Nor do any of the IPCC AR6 SPMs. So the first sentence is not correct: the Paris agreement does not primarily focus on global warming "targets". And it is a mistake to think of warming levels like 1.5C or 2C as targets; just because we do not wish to exceed them does not mean that these warming levels are where we wish the climate to arrive at or remain, either as a peak temperature or after a peak-and-decline. So in this paper, there are basically two distinct usages of the word "target": in the control-theory sense of "setpoint", and in the policy sense of reference warming level, and I advocate that the authors do not conflate them. I.e., there is a 1.5 degree global warming level as specified in the Paris agreement, and because one is interested in both the impacts at that or another policy-relevant global warming level, as well as the emissions pathway that is compatible with that warming level and how it might evolve over time, one uses the AERA-MIP protocol with a setpoint at that level to drive the simulations to that setpoint. I think that it is very important *not* to use the word "target" for the setpoint value, because doing so implies a value judgement that the setpoint warming level is in fact a desired goal. So I would strongly encourage the authors to replace every single instance in the entire manuscript of the word "target" for either of the more neutral words "level" or "setpoint", or other similar wording where appropriate, throughout (including when used within figures such as figure 1).

We thank the reviewer for highlighting this important nuance. In the revised version of the manuscript, including Figure 1, we have replaced the word '*target*' with more neutral terms such as global warming '*level*'.

My second major concern is about how this manuscript treats ACCESS as an outlier. The overall hypothesis of the manuscript seems to be that it is possible to apply the AERA in a set of full-complexity ESMs, but then the one model that does not work as intended is not fully included in the analysis. The implications of this would seem to be pretty important, as it implies that certain types of physical uncertainties might lead to a failure of climate policies that are structured like the AERA approach. The explanation on lines 132-136 makes sense,

but what is the evidence for it? How confident are we that this mismatch in non-CO2 forcing is not pointing to a real uncertainty that we need to consider in remaining emissions budgets? So I would advocate keeping ACCESS within the ensemble statistics unless a stronger case is made that the non-CO2 forcing estimate difference represents an unphysical model artifact.

We note that the ACCESS model also converges to both prescribed temperature levels, but substantially later than all other models. Therefore, we do not fully agree that, according to ACCESS, this would indicate a failure of climate policies. In AERA, an existing mismatch between actual non-CO2 emissions and those supplied to the algorithm is not corrected for. As a policy instrument, scientists would revise initially biased non-CO2 emissions estimates over time, allowing temperatures to converge faster.

We have, however, now included the ACCESS model in a new figure in the Appendix (Figure A3), which depicts the multi-model mean CO2-fe emissions pathways with and without ACCESS. For example in the 1.5ºC scenario, the multi-model mean CO2-fe emissions in year 2050 are 0.03 PgC yr$^{-1}$ with ACCESS and 0.16 PgC yr$^{-1}$ without ACCESS. Between 2100 and 2150, the average CO2-fe emissions are 0.79 PgC yr$^{-1}$ with ACCESS and 1.66 PgC yr$^{-1}$ without ACCESS. Thus, the multi-model mean emissions pathways are nearly identical. We have added the following to the text: "*The inclusion of the ACCESS model in the ensemble statistics may alter our quantitative results, but not qualitatively*".

[Figure]

Figure A3: Simulated multi-model mean CO$_2$-fe emissions for the 1.5°C and 2.0°C warming levels with and without including the ACCESS model.

I am also a bit confused about the implications of removing the historical warming differences between the models by lining them up so that they all pass through 1.22 degrees at year 2020. How was this handled for the models with multiple ensemble members -- were the ensemble members each lined up at their own year 2020 temperature, and thus they have different

absolute temperature setpoints? If so, is there any correlation between the REBs and the absolute temperature at 2020 across the ensemble members? Or was the ensemble-mean global temperature used for the year 2020 value so that all members have the same absolute temperature setpoint? Given that there is real uncertainty in the current level of global warming due to internal variability (e.g. as pointed out on line 218), what are the implications of this choice on the uncertainty in REBs? Some further discussion of this would be helpful.

Yes, each ensemble member was aligned to its own anthropogenic temperature in 2020, resulting in slightly different absolute temperatures for that year. However, the maximum difference between ensemble members in 2020 is very small. For instance, in the 5-member GFDL ensemble simulation, this difference is only 0.067°C. We have added to the text: "*Each ensemble member has its own anthropogenic temperature in year 2020, resulting in very small differences in absolute temperatures for that year (maximum differences across GFDL ESM2M ensemble members of 0.067°C).*"

Regarding the comment about the REB, by definition:

$$REB(t_{st}) = \frac{remaining\ warming\ (t_{st})}{TCRE - fe}$$

with $t_{st}$ the year of the current stocktake, for example 2025 for the first stocktake. The remaining warming at each stocktake is model and simulation dependent. It is defined by the difference between the absolute temperature target and the absolute temperature at the current stocktake: $T_{target} - T(t_{st})$, with $T$ the temperature fit (extended 31-year running mean). In the "relative warming" case presented in this study, the temperature target is defined by: $T_{target} = T(2020) + observed\ remaining\ warming$. The observed remaining warming is 0.28°C for the 1.5°C scenario and 0.78°C for the 2°C scenario (see Methods). Thus, for example for the 1.5°C scenario, the remaining warming is equal to $T(2020) - T(t_{st}) + 0.28$.

On the other hand, the TCRE-fe is defined by the ratio of the temperature anomaly relative to preindustrial over the cumulative emissions: $TCRE - fe = \frac{T(t_{st}) - T_{piC}}{\sum_{1850}^{t_{st}} E_{total-fe}}$.

Which finally gives:

$$REB(t_{st}) = (\ T(2020) - T(t_{st}) + 0.28)\ \frac{\sum_{1850}^{t_{st}} E_{total-fe}}{T(t_{st}) - T_{piC}}$$

Thus, while the REB depends on the absolute temperature in 2020, it also depends on the absolute temperature of the current stocktake, which means we do not find a clear linear relationship between the two across the GFDL ensemble members (see Figure R2).

[Figure]

Figure R2: Scatterplot between simulated anthropogenic temperature in year 2020 and the remaining emissions budget at the first stocktake for the individual ensemble members for both the 1.5°C warming scenario (red dots) and the 2.0°C warming scenario (blue dots). The 2°C scenario consists only four members in this figure, as the metadata from the AERA algorithm was missing for one member.

Lastly, I recommend revising the schematic in figure 1 bottom-right panel to show that the minimum in temperature spread is at present-day, and then increases prior to that, rather than what is shown where the temperature spread increases over time starting from preindustrial conditions.

We decided not to change Figure 1. Its purpose is to highlight the difference between the traditional IPCC forward approach (prescribed emission pathway → temperature response) and the new AERA approach (prescribed temperature level → adaptive emissions pathway). Introducing the concept of the relative temperature target would confuse the general reader.

Why is the time evolution of the ensemble spread of CO2-fe in figure 3b and 3c so different between GFDL-ESM2M and EC-Earth?

The number of ensemble members is very small for both GFDL ESM2M (5 ensemble members) and EC-Earth (3 ensemble members). Therefore, the magnitude of the ensemble spread and its temporal evolution should be interpreted with caution. For example, we cannot determine with confidence if the ensemble spread changes in magnitude over time.

Line 172-175: The E_LUC won't be quite the same as in the references, since the temperature and CO2 pathways will differ, and thus the loss of additional sink capacity and weakening of carbon-climate feedbacks will be different. Can you quantify that effect, and does it matter?

Correct. But we cannot quantify this effect. The difference may not be that large because all models converge relatively rapidly despite this mismatch between the real E_LUC and the E_LUC prescribed to the AERA. No changes are made to the text.

Lines 216-217. Is the +/- 10% correspond to the light red and blue shading in fig. 1? Or is it the +/- 0.2C referred to on line 217? Either way, please note that in the figure caption.

We cannot find the +/-10% statement the reviewer is referring to. In addition, we assume that the reviewer refers to Fig. 2 and not Fig. 1, as blue and red shading is only shown in Fig. 2. We have added to the caption of Figure 2: "The horizontal shading in (a) and (b) indicates the uncertainty with which anthropogenic warming can be determined (± 0.2°C)"

In fig. B3, why is the difference between the AERA-derived and "true" estimate of TCRE so systematic over the early years of the experiment? I.e. the multi-model ensemble looks roughly to have about the same difference as individual models. Is that due to some systematic path dependence in aerosol, land-use, and other SLCF dynamics?

This issue is related to the temperature fit and not due to aerosols or other factors. We explained this in the revised Figure caption: "*Solid lines show TCRE-fe calculated by AERA at each stockake year. Dotted lines indicate the TCRE-fe calculated a posteriori with the final ("true") estimate of the 31-year running mean timeseries of GSAT. The deviation between the solid and dotted lines occurs because the temperature fit becomes less accurate when the temperature increase slows toward stabilization. As the warming rate decreases, the 31-year mean will also decrease. Therefore the "true" estimate of the 31-year mean will be smaller than the estimate at each stocktake. Consequently, the TCRE will be smaller than estimated at the stocktake during periods of slowing down temperature increases. In addition, the reconstructed TCRE-fe (dotted lines) is less variable than the TCRE-fe calculated by AERA, which calculates $\Delta T$ based on the last year of the extended 31-year running mean timeseries until each stocktake year, which introduces some noise compared to the true final estimate.*"